# Computational Studies of Aflatoxin B_1_ (AFB_1_): A Review

**DOI:** 10.3390/toxins15020135

**Published:** 2023-02-07

**Authors:** Joel Martínez, Maricarmen Hernández-Rodríguez, Abraham Méndez-Albores, Guillermo Téllez-Isaías, Elvia Mera Jiménez, María Inés Nicolás-Vázquez, René Miranda Ruvalcaba

**Affiliations:** 1Departamento de Ciencias Químicas, Facultad de Estudios Superiores Cuautitlán, Universidad Nacional Autónoma de México, Cuautitlán Izcalli 54740, Mexico; 2Laboratorio de Cultivo Celular, Escuela Superior de Medicina, IPN, Salvador Díaz Mirón esq. Plan de San Luis s/n, Casco de Santo Tomas, Miguel Hidalgo, Ciudad de Mexico 11340, Mexico; 3Unidad de Investigación Multidisciplinaria L14 (Alimentos, Micotoxinas y Micotoxicosis), Facultad de Estudios Superiores Cuautitlán, Universidad Nacional Autónoma de México, Cuautitlán Izcalli 54714, Mexico; 4Department of Poultry Science, University of Arkansas, Fayetteville, AR 72701, USA

**Keywords:** mycotoxin, AFB_1_, in silico, DFT, docking, molecular dynamics

## Abstract

Aflatoxin B_1_ (AFB_1_) exhibits the most potent mutagenic and carcinogenic activity among aflatoxins. For this reason, AFB_1_ is recognized as a human group 1 carcinogen by the International Agency of Research on Cancer. Consequently, it is essential to determine its properties and behavior in different chemical systems. The chemical properties of AFB_1_ can be explored using computational chemistry, which has been employed complementarily to experimental investigations. The present review includes in silico studies (semiempirical, Hartree–Fock, DFT, molecular docking, and molecular dynamics) conducted from the first computational study in 1974 to the present (2022). This work was performed, considering the following groups: (a) molecular properties of AFB_1_ (structural, energy, solvent effects, ground and the excited state, atomic charges, among others); (b) theoretical investigations of AFB_1_ (degradation, quantification, reactivity, among others); (c) molecular interactions with inorganic compounds (Ag^+^, Zn^2+^, and Mg^2+^); (d) molecular interactions with environmentally compounds (clays); and (e) molecular interactions with biological compounds (DNA, enzymes, cyclodextrins, glucans, among others). Accordingly, in this work, we provide to the stakeholder the knowledge of toxicity of types of AFB_1_-derivatives, the structure–activity relationships manifested by the bonds between AFB_1_ and DNA or proteins, and the types of strategies that have been employed to quantify, detect, and eliminate the AFB_1_ molecule.

## 1. Introduction

Aflatoxins are metabolites of *Aspergillus flavus* Link *ex* Fries, a fungus that under certain conditions, grows prolifically on peanuts, cereals, and milk produced by cows [1,2]. AFB_1_ is the most carcinogenic of the aflatoxins (aflatoxin B_2_ (AFB_2_), aflatoxin G_1_ (AFG_1_), and aflatoxin G_2_ (AFG_2_); aflatoxin B_1_ (AFB_1_, Figure 1) [3]). It is well-documented that the title molecule is the causal agent of hepatocellular carcinoma (HCC) [4]. Consequently, it has been categorized, by the International Agency of Research on Cancer, as a human group 1 carcinogen [5]. It is important to note, that the corresponding isolation and structural elucidation, shown in Figure 1, was reported in 1963 [6]; later in the 1970s’, the corresponding crystal analysis was obtained [7,8].

AFB_1_ is metabolized in the liver by the microsomal cytochrome enzyme (CYP450), resulting in the formation of AFB_1_-*exo* 2,3-epoxide; this compound exhibits genotoxicity and leads to the induction of reactive oxygen production [9]. The immunotoxicity of aflatoxins is probably the second most documented toxicological effect, and it is important to note that its mechanism of action is not still well explained [10]. In general, the immunosuppressive effects of aflatoxins have been demonstrated [11]; however, immunostimulatory actions have been recently reported [12]. Additionally, many other aflatoxin-related pathologies, such as malnutrition diseases, retarded physical and mental maturity, alterations in reproduction, and nervous system diseases, among others, have been demonstrated in humans or animals. The understanding of the molecular mechanism involved in the toxic effects can be addressed by computational studies [13].

Of specific importance for this work, it is appropriate to comment that the advancements in technology have allowed both the birth and development of new branches of science; in this sense, it is crucial to highlight the area of computational chemistry (CC) [14]. Consequently, many studies have been conducted and many research groups have emerged, promoting the corresponding technological advancement, and consequently increasing the expertise in this discipline. CC aims to predict various molecular properties of chemical systems, also offering information to rationalize or interpret trends and structure–activity relationships [15,16]. In addition, CC employs a wide range of theoretical techniques. Since computational chemistry is suitable for understanding chemical systems, it has been applied as a learning and support tool in experimental investigations. CC comprises two main methods: molecular mechanics and quantum mechanics [17,18,19,20]. The latter is included within the theoretical models and is based on the Erwin Schrödinger equation; in this sense, the method used depends on the nature of the molecular model and the requirements. Thus, basic knowledge of the foundations of the theoretical methods, the ability to critically analyze the results, and some skills in handling software and hardware are necessary. In particular, quantum chemistry (QC) describes the spatial distribution of electrons and determines molecular properties; this field employs several electronic structure methods, such as semiempirical, Hartree–Fock, and density functional theory (DFT), which have been utilized for the description of matter in terms of electrons and nuclei. The theoretical results depend on the level and basis set used in the calculations, making DFT the most popular method in QC [21,22,23,24]. It is important to note that the computational methodologies, such as DFT calculation and molecular mechanics (docking and molecular dynamics), have the advantage of reproducing experimental data while requiring less computing time and memory.

It is also convenient to comment that the theoretical determinations have been accomplished with the purpose to obtain an explanation of the chemical properties of a target molecule, such as their geometrical, electronic and spectroscopy; this in addition to toxicity as the interaction with metals and mechanism of action of AFB_1_ [25,26,27].

As a part of our research program, in CC, we have performed valuable studies contributing to knowledge of the chemical behavior of AFB_1_. This information can be conveniently summarized as follows: (a) a mass spectrometry/mass spectrometry study on the degradation of AFB_1_ in the maize with aqueous citric acid [28]; (b) theoretical calculations to confirm that the active site corresponds to the lactonic ring [29]; (c) a theoretical study [30] of 8-chloro-9-hydroxy-AFB_1_, which determined the structural, electronic, and spectroscopic properties of this reaction product of AFB_1_; (d) a theoretical study [31] related to the adsorption process of B-aflatoxins using a vegetable specimen *Pyracantha koidzumii* (Hayata); (e) in a recent publication [32], the effectiveness of lettuce and field horsetail as biosorbents for the elimination of aflatoxin AFB_1_; was evaluated (f) concluding with a recent in silico work [33], describing the interaction of chlorophyll with AFB_1_.

Considering the previous arguments, the goal of this work is the accomplishment of a review, compiling conveniently all the contributions encountered after an extensive search of the literature, related to studies of AFB_1_ by CC.

It is important to highlight that to achieve an appropriated order, the manuscript was organized into five sections. In the first instance, structural, energy, solvent effect, ground and the excited state, atomic charges, Wiber bond indices, bond order value, vibrational frequencies, thermodynamics parameters, among others of AFB_1_ are presented. Then, several theoretical investigations about the reactivity of AFB_1_. Afterward, the revisions related to the molecular interactions of AFB_1_ with inorganic compounds; then, molecular interactions with environmentally compounds; and, finally, the research related to the molecular interactions with DNA, enzymes, cyclodextrins, glucans, among others biological compounds.

## 2. Methods

It is also convenient to highlight that the literature search was accomplished by employing the SciFinder^®^, Scopus, Google Scholar, and Researchgate databases, considering the following keywords:Aflatoxin B_1_Aflatoxin B_1_ quantum chemistryAflatoxin B_1_ computational studiesIn silico studies of aflatoxin B_1_In silico studies of AFB_1_DFT studies of AFB_1_Docking studies of AFB_1_Molecular dynamics studies of AFB_1_

From a deep search, approximately 3000 references were found; however, only were selected works related to our interest topic (AFB_1_).

## 3. Results and Discussion

### 3.1. Molecular Properties of AFB_1_ (Structural, Energy, Solvent Effect, Ground and the Excited State, Atomic Charges, among Others)

To date, semi-empirical methods are still used, and acceptable results are obtained for certain properties. According to the technological advance of computers, ab initio methods (Hartree–Fock) are also used. Some conventional calculations of molecular properties are based on the description of the individual motion of electrons. In ab initio methods, it is not necessary to consider the individual movement of each electron; it is enough to know the average number of electrons located at any point in space, and this gave rise to the Density Functional Theory (DFT) method. The simplicity of this methodology has made it possible to study complex molecules. Once the optimized molecule is obtained, electronic properties such as geometric and spectroscopic parameters, electronic density, electronic energy, and thermochemical and thermodynamic properties, among others, can be determined to explain the chemical or reactive behavior of molecules.

According to the above methodologies, in this section are explained the electron affinity, carbocation formation, behavior as an electrophilic species, ^13^C chemical shifts, oxidation, and AFB_1_ interaction site; additionally, also AFB_1_ is unstable in the presence of external fields.

(i). Heathcote and Hibbert [34] determined the electronic structure of AFB_1_ and its 4,19-dehydro derivative by the simple Hückel molecular orbital method to determine their structure–activity relationship. Hence, the molecular orbital calculations were expressed in terms of the bond order of the π-electrons. The comparative results of the corresponding molecular orbital calculations showed that the 2,3-π bond has the highest bond order value (0.9561). Consequently, the AFB_1_ appeared as the most reactive molecule in comparison to its 4,19-dehydro derivative, since the 2,3-π bond is isolated from the conjugated system of the molecule, diminishing the reactivity of 2,3-π bond (0.8299). The authors pointed out that simple Hückel approximation of the molecular orbital method is appropriate to determine the relative electron affinities of aflatoxin.

(ii). In this work [35], Pachter and Stey performed several Semi–empirical Intermediate Neglect of Differential Overlap (INDO) calculations to obtain both the atomic charges (*q*) and Wiberg bond indices *p* (in electrons, e^−^) for a set of three molecules, AFB_1_, sterigmatocystin (ST), and versicolorin A (VA). The attained results were analyzed, establishing a possible structure–activity relationship for AFB_1_ and its biosynthetic precursors; see Table 1. Bearing in mind that the AFB_1_ epoxide formed by the oxidation of the vinyl ether bond of AFB_1_ has been proposed as the active carcinogenic metabolite, this statement was supported by the bond order findings for AFB_1_, ST, and VA. A relatively large negative charge was detected at the C4 carbon atom of AFB_1_, whereas no charge polarization was noted in ST and VA. In this sense, the partial atomic charges of the furan ring in AFB_1_, ST, and VA were −0.26, +0.26, and +0.27 electrons, respectively. An interesting conclusion related to the obtained results can be stated as “the higher electron density at the C2 carbon atom, in AFB_1_, implicates a higher stability of the AFB_1_ carbocation (formed from the corresponding epoxide) in comparison to ST and VA, accounting for the higher biological activity of AFB_1_”.

(iii). In this research, the Austin model 1 (AM1) was employed to obtain the fully optimized geometry of the aflatoxins B_1_, B_2_, G_1_, and G_2_, in addition to their corresponding heats of formation [36]. AFB_1_ (Figure 2) is largely a planar molecule.

As it can be seen, the heavy atoms except C2, C3, and O1, remain around the same plane, and the outer dihydro-furan moiety does not project significantly from this plane. As expected, slight variations between the calculated interatomic distances of AFB_1_ and the experimental values of X-ray diffraction were revealed. In this sense, the highest bond order of the AFB_1_ corresponded to the C2-C3 bond; it is important to note that these atoms are involved in both epoxidation and hydroxylation. Afterwards, in a comparison between AFB_1_, the AFB_1_ epoxide, the AFB_1_OH, and the AFB_1_ radical, several significant variations were observed, both in the atomic charges and the bond indices around C2 and C3; thus, the atomic charge of C3 was from −0.20 in AFB_1_ to near zero in the AFB_1_ epoxide and AFB_1_OH. However, the atomic charge of C2 was close to zero; this fact can be attributed to the radical formation by removing the hydrogen atom bonded to C2.

Furthermore, the calculated heat of formation, LUMO (lowest unoccupied molecular orbital), and HOMO (highest occupied molecular orbital) energies of the aflatoxins B_1_, B_2_, G_1_, G_2_, and AFB_1_ derivatives are summarized in Table 2. The corresponding data are indicative of the following findings: the AFB_1_ epoxide is thermodynamically more stable than the AFB_1_ by 29 kcal/mol, and AFB_1_OH is almost twice as stable as the corresponding epoxide heat of formation. The assessed low LUMO energies indicate that the aflatoxins are electrophilic species. In fact, the calculated low LUMO energies indicate that the aflatoxins are electrophilic species.

(iv). Barone et al. examined the geometry optimization and GIAO (gauge including atomic orbitals) ^13^C NMR chemical shift calculations at the Hartree–Fock (HF) level, using the 6-31G(d) basis set, were projected as a tool to achieve the structural characterization of every organic compound, providing suitable support in the interpretation of experimental NMR data [37]. Hence, in this work, Barone et al. presented the parameters related to the linear correlation plots of computed versus experimental ^13^C NMR chemical shifts for fourteen low-polar natural products containing 10–20 carbon atoms, highlighting the case of AFB_1_, at the HF level, that has a r-value of approximately 0.995 which indicates an excellent method. Moreover, a comparison with the hybrid B3LYP method was conducted to assess the electron correlation contributions to the evaluation of the ^13^C NMR chemical shifts and to extend the applicability of such computational methods to the interpretation of NMR spectra in apolar solutions. The comparison was tested by studying three examples of revised structure, previously assigned, examining how the theoretical ^13^C chemical shifts matched the experimental data.

(v). The Gaussian program package was employed by Billes et al. [38], with the DFT B3LYP functional and the 6-31G* basis set. According to the obtained calculations, the four rings (B–E)—see Figure 1—of the AFB_1_ are coplanar, independently of the saturated or unsaturated state of the ring A. Some changes were perceived via the demethylation of the O22-C23 bond: the OH bond was generated, in addition to minor changes in the C15-C14 distance, and for the C16-C15-O22 and C14-C15-O22 valence angles. The demethylation of the O22 atom increases the positive charge on C15. Moreover, and as expected, the change from CH_3_ to H increases the polarity of the OR groups. The charge of the O22 atom transforms into a more negative value, and the H23 atom retains a positive charge in comparison to the negative data of the C23 atom. Some other results related to the corresponding spectrophotometric infrared absorption were also acquired, namely the vibrational, deformational and torsional force constants, and the assignments of the spectral bands; in this sense, the stretching force constants were unaffected by demethylation. The calculated quantities reflect the changes in the molecular structure. The Gibbs free energy of the demethylation reactions, in addition to their equilibrium constants, was calculated. Related to the achieved reaction products, the equilibrium constants offered the order of methanol and formaldehyde production; these results explain the role of methanol formed and its oxidation product in the mechanism behind the toxic action of aflatoxin.

(vi). Li et al. [39], reported a computational study of the AFB_1_ isomers using DFT/B3LYP/6-311+G(d,p). Through-single point calculations and geometry analysis, it is shown that a *cis*-relation is more stable than a *trans*-relation; see Figure 3. It is convenient to note that the Raman spectra of both geometric isomers (*cis* and *trans*) were calculated and compared with the AFB_1_ *cis*-experimental Raman spectra, finding appropriate data correlation. Moreover, a contribution to the frontier orbitals was evaluated, establishing that the electrophilic ability of molecules was stronger than the nucleophilic ability. For C13, LUMO orbitals contribute 21.48 and 20.62 percent in the *cis* and *trans* isomers, respectively. It is important to highlight, that the obtained data suggest that the C13 atom can serve as a position for an interaction with DNA. Li et al. proposed that the research has direct theoretical relevance to *cis*-*trans* isomers’ detection, their transformation, and the inhibition of toxicity.

(vii). The DFT/B3LYP level and 6-311G basis set were applied to investigate the effect of external fields on AFB_1_ by Junbo et al. [40]. The total energy, dipole moments, geometric parameters, energy gap, and spectrophotometric infrared absorption under different external fields, from 0 to 0.015 a.u. (atomic units at the electric field), were analyzed. Moreover, the UV–Vis spectral absorption of AFB_1_ and excitation states were calculated with the time-dependent DFT method. An increase in the external electric field promotes a gradual improvement in the total energy of the molecules; however, the dipole moment steadily decreases. Molecular geometric parameters are strongly dependent on the increasing field intensity. The bond length of C6-O7 and C15-C14, among others, decreases with the increase in the external electric field, and the bond lengths of C16-C15 and C8=O20 show a trend of first decreasing and then increasing. In addition, the energy gap revealed an evident decreasing trend, indicating that AFB_1_ is more easily excited by the external electric field, provoking it to react chemically.

The obtained vibration modes in the infrared spectrum caused different changes, and the two strongest vibration modes occurred on the benzene ring. The stretching vibrations of C15-C14, C5-C17, and in both C=O groups (F < 0.0025 a.u.) corresponded to a blue shift (Figure 4). The AFB_1_ molecule displayed two absorption peaks, 193.0 and 322.0 nm, in the absence of an external electric field. However, under the presence of an external electric field, the absorption peak at 193.0 nm was red-shifted, and at the absorption peak at 322.0 nm, a blue shift occurred, although the total absorption line was still in the near-ultraviolet region. Finally, the authors recognized that during an excitation energy decrease, the AFB_1_ molecule is unstable under external fields.

### 3.2. Theoretical Investigations of AFB_1_ (Degradation, Quantification, Reactivity, among Others)

As it was previously stated, using electronic structure, in Section 3.1, in this section are included some molecular simulation methods to analyze the reactivity aspects such an *endo*-attack, a stable ligand-DNA adduct, and electronic excitation to explain a transition or charge transference, interaction energies, *exo*-2,3-epoxide formation, the interaction between protonated AFB_1_ and carboxylic groups, reactive species; and conditions for inactivation and mitigation of producer of AFB_1_.

(i). In this research, Okajima and Hashikawa [41], an ab initio molecular orbital (MO) calculation was performed to explore the solvent effect during the SN_2_–type nucleophilic oxirane ring opening of AFB_1_ epoxide, considering the *endo* and *exo* stereoisomers I and II respectively (Figure 5). The evaluated reactions were accomplished considering I and II interacting with NH_3_ as a nucleophile. In this case, (H_2_O)_n_ (n = 1–3) was the interaction solvent with the oxirane oxygen; in addition, the evaluated stationary points, including transition structures (TSs), were optimized with no geometrical constraint in the restricted Hartree–Fock (RHF)/3-21G basis set; moreover, the relative energies were estimated at the B3LYP/3-21G (Becke (3) Lee, Yang, Parr) level based on the RHF/3-21G geometries. The obtained results are summarized as follows. (1) The (H_2_O)_1–3_ systems are associated with the oxirane oxygen for the reaction of I, II; however, steric repulsion between the H_2_O molecules and AFB_1_ unit during the incoming of the nucleophile is attained; consequently, a higher reactivity of the *exo* isomer with the AFB_1_ epoxide was detected, well defined by the difference in solvent effect according to the evaluated TSs. (2) For the reaction of II, the approach of H_2_O molecules to associate with the oxirane oxygen can occur from three directions (outside, backside, and frontside) considering the stereostructure of AFB_1_. The involved energy of one and two associated H_2_O molecules, stabilizes the ability of TS, in the following order: outside > backside > frontside. This implies that the steric repulsion between H_2_O and the AFB_1_ unit is in the order of outside < backside < frontside. (3) The energy difference between the most stable *exo-* and *endo*-attacking TS, increases according to the major number (n = 0–3) of associated H_2_O molecules. In conclusion, the authors state that the solvent effect implies an *endo*-attack of the nucleophiles predominantly.

(ii). In the following study, Okajima et al. performed theoretical calculations to obtain the binding mechanism of intercalators to DNA base pairs [42]. According to the results, the structures of the intercalating complexes and ligand–DNA adducts of the mutagenic aflatoxin β1-2,3-oxide (**1**) and benzo[a]-pyrene diol oxide (**2**) and their reactivity for a type-SN_2_ oxirane ring opening for (**2**) were explored (Figure 6). It is important to highlight that Okajima et al. concluded that the stability of ligand–DNA complexes and adducts, rather than the reactivity for the SN_2_ reaction between two reactive sites, plays an essential role in carcinogenic or mutagenic potency.

However, it is convenient to consider the findings achieved in this study: the interactions between two electron lone pairs of nitrogen of the guanine residue in DNA, N2 or N7, and an electron-deficient carbon of the oxirane ring of the mutagenic substrates (**1**) and (**2**) are considerably slight, favoring the promotion of the covalent binding of (**1**) and (**2**) to guanine. For the adducts of (**1**) and (**2**) with the DNA double helix, an outsized planar coumarin ring of (**1**) exists in the major channel of the double helix, in addition to a planar aromatic ring system of (**2**) in the corresponding minor groove; hence, the repeatedly layered structure of the B-type DNA double helix is not distorted in the respective adduct of (**l**). Nevertheless, the double helix is extensively distorted in the adduct of (**2**). Finally, from the obtained activating energy data, i.e., 47.7 kcal/mol, the reactivity for covalent binding is noticeably small, suggesting that intercalation must play a significant role as a phenomenon preceding the covalent binding between the nucleophilic (guanine) and the electrophilic centers of the mutagens. In other words, the potency for mutagenesis could be linked to the development of a stable, long-standing intercalating complex and stable ligand–DNA adduct.

Complementarily, the geometry optimizations of the model ligand–DNA complexes of (**1**) and (**2**) were performed using the Macromodel program (ver. 4.5) with the MM2 parameter. The activating energies were estimated by the following equation: ΔE = ETS − E(**1** or **2**) + EG-C. Energy evaluations were performed employing benzo[*a*]-pyrene oxide without two OH groups.

(iii). Guedes and Eriksson [43], conducted a theoretical study of the main structural, electronic, and photochemical properties (phototoxic reactions) for a set of aflatoxins at the B3LYP/6-31+G(d,p) level, to understand their molecular behavior. In this sense, the results suggest the possibility of aflatoxins to generate reactive oxygen species (ROS), superoxide anions, hydroxyl radicals, and singlet oxygen. Moreover, the higher toxicity of AFB_1_ was discussed in terms of its lower ionization potential. In addition, using molecular orbital determinations, Guedes and Eriksson observed that the oxygen-dependent photochemical reactions involve two pathways: (a) electronic excitation to the first excited singlet state AFB_1_(S_0_) *h*ν → AFB_1_(S_1_); (b) intersystem crossing to the first excited triplet state, AFB_1_(S_1_) ISC → AFB_1_(T_1_). Both occur by eventual reduction and electron transfer, or the direct transfer of the triplet excitation energy, toward molecular oxygen (Figure 7). It is also convenient to comment that for aflatoxin B_1_, the lowest singlet excitation occurs at 378 nm, and the third singlet excitation, occurring at 3.96 eV, showed the highest probability (0.29), representing predominantly a HOMO (−2)–LUMO transition.

(iv). Méndez-Albores et al. [28], proposed that the reaction between citric acid dissolution and AFB_1_ activates the hydrolysis of the lactonic ring in the target molecule, with results supported by means of tandem mass spectrometry (MS/MS) and computational data. Related to the two main products achieved during the acidification process, they were recognized by the corresponding mass spectral data; a peak *m*/*z* 286 correspond to the lactone group and the fragment-ion *m*/*z* 206 was proposed for the cyclopentenone ring of the AFB_1_. It is important to highlight that these fragments -ions, fit to the corresponding molecular ions, which consequently agree with their respective molecular weights. In line with the theoretical calculations, applying density functional theory with B3LYP/6-31G(d,p) level, was established that the active site corresponds to the lactonic moiety. On the other hand, considering the fluorescent feature of the target molecule, the results agree with the fact that the rings A, B, C, and D of the AFB_1_ adopt a planar conformation, with all dihedral angles varying by less than one degree from planarity, whereas ring E is located slightly outside the plane, enabling the formation of an extended conjugated π–electron system. Moreover, the charge values obtained using the natural population analysis demonstrated an electronic deficiency in the carbonylic carbon atom, favoring this site for a nucleophilic attack, providing the driving force to hydrolyze the lactonic moiety. The charge transference of the lactone ring and of some carbon atoms of the benzene ring highlighted the existence of a conjugation among them; hence, the charge transfer observed between the ground and the excited singlet state disclosed by the fluorescence was supported by an increase and a decrease in the electronic charge of the atoms involved in the lactone ring. Therefore, the fluorescence phenomenon diminishes when the aflatoxin is hydrolyzed.

(v). In this work, Nicolas-Vazquez et al. reported a study employing quantum chemistry at the DFT level, explaining the chemical behavior of the lactone ring present in AFB_1_, under acidic hydrolysis [29]. In this sense, the geometrical and electronic parameters were employed to suggest reactivity, using the B3LYP/6-311+G(d,p) method. The obtained results indicated that the fused A, B, C, and D rings adopt a continuous planar conformation; moreover, the carbon atom, C8 (Figure 1), of the lactone group presented a highly electrophilic character, since the population analysis yielded a high positive charge for this atom, 0.779 e^−^, whereas high negative charges were recorded for both oxygens, O20 and O7, of −0.528 e^−^ and −0.548 e^−^, respectively. In addition, the atomic charges, electrostatic potential, and HOMO and LUMO indicated that the oxygen atom O20 is the more suitable site for the aflatoxin moiety for interaction with protons generated by the aqueous citric acid conditions. Thus, in an acidic aqueous medium, the oxygen atoms must be protonated, favoring the carbon site to be nucleophilically attacked by water. Accordingly, the OC8-O7 bond length has been lengthened substantially, to 1.462 Å. It was also demonstrated that the lactone ring of AFB_1_ is hydrolyzed under acidic conditions by an acid-acyl bimolecular mechanism, A_AC_2, suggesting the deletion of its carcinogenic properties.

(vi). The paper of Wyszomirski and Prus [44] commences by highlighting that the contamination of food and drinking water with toxic substances is a serious problem nowadays. Additionally, due to the low concentration of AFB_1_, its quantification is a severe problem; thus, in the paper, it is also mentioned that in order to improve AFB_1_ quantification, the preparation of molecularly imprinted polymers (MIPs) is an appropriate strategy to mimic the properties of antibodies; it is additionally cited that the preparation of MIPs involves the polymerization of functional monomers in the presence of template molecules, cross-linking monomers, initiators, and a solvent (named grafting solution). Thus, Wyszomirsk and Prus in 2012 [vide supra] selected a computational model using the Materials Studio package to determinate the lowest energy conformer, to identify the interaction energy of AFB_1_ and 5,7-dimethoxycoumarin (DMC) as templates. In addition, four monomers were considered: (allylamine, methacrylic acid (MMA), 2-(diethylamino) ethylmethacrylate (DEAEM), and *N*,*N*′-methylene *bis*-acrylamide (MBA)) and toluene as a solvent to carry out the grafting solution. Hence, after polymerization, the template molecules were removed, leaving accessible binding sites and the spatial configuration of the target molecule. In this sense, the best interaction energies for DMC and AFB_1_ were obtained for allylamine (−46.4 and −43.1 kcal/mol, respectively), and methacrylic acid (−33.6 and −32.0 kcal/mol, respectively). It is worth noting that both monomers offered similar binding for AFB_1_ and DMC using the MIP method; in addition, a higher number of hydrogen bonds was noted: five hydrogens bonds were appreciated for AFB_1_–MMA and DMC–MMA complexes, with four and six hydrogen bonds with AFB_1_ and DMC, respectively, in the presence of allylamine. In addition, a set of interaction energy values for the grafting solution were established: DMC–MMA and AFB_1_–MMA, −3.9 and −3.8 kcal/mol, respectively; and −5.8 and −5.5 kcal/mol for DMC–allylamine and AFB_1_–allylamine, respectively.

(vii). Karabulut et al. described the selective reduction of the carbonyls in AFB_1_ achieving aflatoxicol (AFL)—see Figure 8—and its molecular structure in addition to the effects on toxicity, was analyzed [45]. The B3LYP/6-31+G(d,p) and B3LYP/6-311++G(2d,2p) levels of theory were applied for geometry optimization, frequency, and solvent effect (water) calculations; complementarily, electrostatic potentials and molecular orbitals were also calculated for AFB_1_ and AFL, comparing the reactivity of the furofuran ring double bond, which causes the formation of an *exo*-epoxide analog (AFB_1_ epoxide).

AFL, a metabolite of AFB_1_, is produced by the selective reduction of the carbonyl group of the cyclopentanone moiety of AFB_1_; this chemical-system implicates two stereoisomers, AFL_1_ and AFL_2_, since a stereogenic center is generated at C10.

Related to the above commentary, the obtained results are summarized as follows. The AFL displayed two possible tautomers (TAFL and TTAFL), and each tautomer has one stereoisomer (Figure 8); the AFL_1_ tautomer was the most stable, both in the gas and water phases. The electrostatic potential energy (ESP) maps obtained for the AFL tautomers (Figure 9) support this explanation; these ESP maps of AFB_1_ and AFL are similar, especially around the 2,3-double bond on the furan ring. This theoretical result supports the recent experimental data, which consider AFL to be as carcinogenic as AFB_1_. Moreover, the similarity in the distribution of HOMO and LUMO orbitals on the furan ring is remarkable, and the contribution of the furan ring to the HOMO is almost the same in both structures, through the double bond and the oxygen. Furthermore, the epoxide formation is directly related to the reactivity of the double bond in the furan ring, and according to the calculated ESP maps and orbital diagrams, AFB_1_ and AFL have almost the same epoxide formation tendency, as measured by the double bond reactivity. Finally, according to the theoretical results, TAFL and TTAFL structures can generate *exo*-2,3-epoxide and may act as toxic and carcinogenic molecules, similarly to AFB_1_ and AFL.

(viii). A theoretical study to explain the formation of 8-chloro-9-hydroxy-aflatoxin B_1_ in the detoxification process of AFB_1_ with neutral electrolyzed water was conducted in 2016 [30]. This molecule (8-chloro-9-hydroxy-aflatoxin B_1_) is the reaction product of AFB_1_ with hypochlorous acid (HOCl) from neutral electrolyzed water. Sixteen isomers (stereoisomers and regioisomers) of the molecule were optimized using the DFT (B3LYP/6311++G(d,p)) level. However, the most stable isomer was with the chlorine in the *anti*-position to the hydroxyl group. In this work, two reaction pathways were assessed: ionic species (Cl^+^ and OH^−^) and the entire hypochlorous acid molecule. In general, the authors proposed that the reaction included an electrophilic attack by the double bond of the AFB_1_ molecule to the chlorenium ion (ionic), or hypochlorous acid (molecular) to produce a chloronium ion reactive intermediate. Subsequently, the hydroxide ion attained a nucleophilic attack on the intermediate, forming 8-chloro-9-hydroxy-aflatoxin B_1_.

Additionally, some toxicological properties of the compound were predicted using the OSIRIS-Property-Explorer software. In general, the 8-chloro-9-hydroxy-aflatoxin B_1_ molecule did not present a risk of mutagenicity but displayed a high tumorigenicity risk and high irritation and reproductive effects. Moreover, the compound showed poor permeability and high solubility, as estimated by its clogP and log S values, respectively. Thus, Escobedo-González et al. concluded that 8-chloro-9-hydroxy-aflatoxin B_1_ has significant absorption but high urinary excretion.

(ix). A study performed by Bonomo et al. in 2017 [46] aimed to understand the reactivity and accessibility of AFB_1_ at all sites of metabolism in CYP3A4 and CYP1A2 by DFT, docking studies, molecular dynamics simulations and free energy (MM/GBSA) calculations. It showed that aliphatic hydroxylation on positions 4 and 12α on AFB_1_ (Figure 1) is energetically favored, whereas position 3 is the preferred site for epoxidation. For this purpose, a series of molecular studies were performed, elucidating the accessibility of each site of metabolism. The results showed that the most stable binding modes in CYP3A4 favor the formation of the 12α-hydroxylated and *exo*-2,3-epoxide metabolites. Conversion of the methoxy group is also sterically possible but not observed experimentally due to its low reactivity. In the CYP1A2 active site, AFB_1_ cannot orient position 12 towards the catalytic center, whereas the *exo*-2,3-epoxide and 4-hydroxylated metabolites are formed from the most stable and the *endo*-2,3-epoxide from a less stable binding mode, respectively.

(x). Using theoretical calculations with DFT (B3LYP/6-311++G(d,p)), Méndez-Albores et al., 2020 [31] evaluated the interaction of AFB_1_ with the functional groups present on the surface of a plant-based (*Pyracantha koidzumii*) biosorbent. In the study, methanol, methylammonium, acetate, and acetone were used as representatives of the hydroxyl, amino, carboxyl, and carbonyl groups, respectively. The best interaction of the AFB_1_ molecule was attained with the carboxylic groups (−40.2 kcal/mol), followed by hydroxyl (−12.8 kcal/mol), carbonyl (−11.4 kcal/mol), and amino (−8.6 kcal/mol) groups. Furthermore, the interactions were also analyzed using theoretical infrared spectrophotometric calculations. Infrared information confirmed the feasibility of the interaction between the protonated AFB_1_ molecule and the carboxylic groups since the band associated with the hydroxyl group showed a significant shift to lower wavenumbers (3436 cm^−1^). Consequently, the authors concluded that carboxylate ion-enriched biosorbents would be the best option for AFB_1_ adsorption under slightly acidic (pH 5) conditions.

(xi). An interesting study related to the food field involved the interactions between several reactive oxygen species (ROS), produced by cold atmospheric plasmas, and AFB_1_. In this context, Li et al. [47] employed reactive molecular dynamics simulation (RMD) with the reactive force field potential, ReaxFF. Additionally, the COMPASS (condensed phase optimized molecular potentials for atomistic simulation studies) II force field was employed. In this sense, the degradation of AFB_1_ (Figure 10), using the cold atmospheric plasma processing of mycotoxin–contaminated food approach, was induced by the addition of ROS (O atoms, OH radicals, and H_2_O_2_ molecules) to the double bond C2-C3, opening ring A, and by the dissociation of the lactone ring. In the first instance (Figure 10a), the addition of the OH radical to C3 was displayed; as indicated, an unstable intermediate (I) was produced, and then the addition of another OH radical to C2 was conducted. The next step involved the oxidation of the C2-C3 bond by the abstraction of hydrogen atoms (Figure 10b). Further, the opening of the furan ring was proposed to occur by two pathways: (1) by the breaking of C19-O1 and C4-C3, and (2) by the rupture of the C2-C3 bond with the formation of two aldehyde groups and cleavage of ring A (Figure 10c). Related to the breaking of ring A (Figure 10d), this is produced by the abstraction of a hydrogen atom supported at C4, mediated by an oxygen atom. In this sense, the C19-O1 bond is separated, and a second subtraction of the hydrogen atom, at C2, generates the intermediate II. Then, the hydrogen supported at C3 is removed by an oxygen atom, generating a OH radical that was incorporated into C3, producing the alcohol group (intermediate III). Then, the hydrogen atom of this group is removed, forming the intermediate IV. Additionally, related to the dissociation of ring D (Figure 10e), this proceeds by two pathways: the first is the reduction of double bond C9-C13 and carbonyl bond C10-O21 (green circles), creating a new double bond between C9 and C10 (blue circle) this process is initiated by an oxygen atom with the abstraction of a hydrogen atom at C12 (red circle), consequently modifying ring E. Furthermore, the reduction of C9-C13 can be initiated by H_2_O_2_. In this case, the H_2_O_2_ was converted into the HO_2_ radical. Hence, the hydrogen atom is abstracted by O21, reducing the C10-O21 bond, and the aromaticity of the benzene ring is broken with the formation of two new double bonds, C13-C14 and C9-C13. In the second case, the rupture of ring D is mediated by H_2_O_2_ (Figure 10f), starting with the elongation (3.273 Å) and the breaking of the O7-C8 bond (intermediate V), and the breaking of the double bond C8-O20, intermediate VI, achieving finally the intermediate VII.

(xii). The inhibition and mitigation of AFB_1_ using pulsed electric fields (PEF) in red pepper flakes was described by Akdemir Evrendilek et al. [48] to determine the inactivation of *Aspergillus parasiticus*, a producer of AFB_1_, the mitigation of aflatoxins, and its mutagenicity. Thus, the prediction of the inactivation and mitigation was performed employing machine learning algorithms (by gradient boosting regression tree model (GBRT-M), and a random forest regression model (RFR-M)), in addition to artificial neuronal networks (ANN), both estimated by the RMSE (root mean square error) and the coefficient of determination value (R^2^). It is important to note that these approaches are able to substitute the statistical method. In this sense, related to the inactivation of *A. parasiticus*, the GBRT-M R^2^ value (0.97) was higher than the RFR-M and ANN values of 0.94 and 0.93, respectively. Related to the inhibition of *A. parasiticus*, the ANN model showed the best values, 0.80 and 0.99, for RMSE and R^2^. Regarding the mitigation of AFB_1_, the obtained data revealed that the ANN model displayed the highest value (95) for R^2^, with a value of 155.20 for RMSE. Finally, these models were compared to predict the optimal PEF treatment conditions, obtaining the best inactivation and mitigation. In this case, the ANN model was the better approach; consequently, this model could be introduced at an industrial scale.

### 3.3. Molecular Interactions with Inorganic Compounds (Ag^+^, Zn^2+^, and Mg^2+^ Ions)

In this section, quantum and molecular mechanic methodologies were used to explain the formation of complexes between AFB_1_ and metal ions, which could indicate how to identify, reduce its toxicity, or eliminate the target compound, and corroborate the stability of these complexes through properties such as dipole moments.

(i). Wu et al. studied a set of high-quality surface-enhanced Raman scattering (SERS) spectra of AFB_1_ and other aflatoxins, using silver nanorod (AgNR) array substrates [49]. In this regard, the authors demonstrated the opportunity to utilize the SERS detection system as a susceptible mycotoxin detection platform. A direct comparison of the experimental (SERS) and theoretical (DFT) Raman spectra of the aflatoxins, revealed an appropriated correlation with the AFB_1_ spectral data. Comparing the DFT-calculated Raman spectra of the studied aflatoxins, several peaks in the Raman spectra of the aflatoxin–Ag complexes were red-shifted; this finding was explained considering the interactions between the aflatoxins through the O atom at the pyran ring with the cyclopentene system and the Ag atom.

(ii) As a complementary study, the structures obtained by SERS and pre-resonance Raman spectra (SERRS) of aflatoxin B_1_ (AFB_1_)-Ag_n_ (n = 2, 4, 6) complexes—see Figure 11—were calculated using DFT with the B3LYP/6-311G(d,p) (C, H, O)/LanL2DZ (Los Alamos National Laboratory 2 Double Zeta) (Ag) basis set [50]. The obtained results indicated that the SERS enhancement factors were between 10^2^ and 10^3^ orders for the AFB_1_-Ag_n_ (n = 2, 4, 6) complexes, respectively, due to the C=O stretching of the pyran ring and the larger static polarizability of the three complexes. The SERRS spectra of the three complexes were obtained by excitation at 407.5, 446.2, and 411.2 nm, determined by time-dependent density functional theory (TDDFT). The SERRS enhancement factors caused by the charge transfer excitation resonance were approximately 10^4^ orders, which corresponds to the Ag-O stretching.

(iii). Gao et al. employed [51], a DFT/B3LYP/6-311G(d,p)/Lanl2dz basis set to investigate the structure, electrostatic properties, and Raman spectra of the AFB_1_–Ag complex. The obtained results suggest that the SERS and pre-resonance Raman spectra of the AFB_1_–Ag complex strongly depend on the adsorption site and the excitation wavelength of the incident light. The SERS factors were enhanced at 10^2^−10^3^ orders in comparison to the typical Raman spectrum of the AFB_1_ molecule, a fact explained by considering the larger static polarizabilities of the AFB_1_–Ag complex. The geometrical structure and the adsorption energy showed that the *a* site of the AFB_1_ molecule is a more favorable adsorption site than the *b* site, Figure 12. However, when the AFB_1_ molecule is adsorbed on a silver nanoparticle by the *a* site, the enhancement factor of the AFB_1_–Ag complex ensued up to 10^3^ orders, in comparison to the typical Raman spectrum of the isolated AFB_1_, due to the chemical environment modification in the AFB_1_–Ag complex. The pre-resonance Raman spectra of AFB_1_–Ag complex were investigated at 266, 482, 785, and 1064 nm incident light wavelengths, resulting in enhancement factors over 10^2^−10^4^ orders, caused by the charge transfer excitation resonance enhancement.

(iv). Altunay et al. designed [52] a ternary complex between AFB_1_, zinc (II), and fluorescein; see Figure 13. The quantum chemical parameters were used to predict the reaction mechanisms of interactions and binding regions of molecules; hence, the optimized geometries and electronic structures of fluorescein, AFB_1_, and their complexes with the Zn (II) ion were calculated using DFT. A hybrid B3LYP functional was used, as well as a multi-basis set built from two standard electronic basis sets: 6-31G(d)_1dz for Zn ions and 6-311G(d)+ for organic ligands. To simulate ethanol as the solvent, a conductor-like screening model (COSMO) with a dielectric constant (ε) = 25.3 was employed. Polarizable continuum model (PCM) radii of hydrogen, carbon, oxygen, and zinc atoms were chosen, at 1.320, 2.040, 1.824, and 1.668 Å, respectively. Energies of complexation Ec for Zn (II)–fluorescein complexes were also calculated as Ec = E(fluorescein) + E(zinc ion) − E(complex). The value of Ec for Zn (II)–fluorescein–AFB_1_ complex was estimated as Ec = E(fluorescein) + E(AFB_1_) + E(zinc ion) − E(complex). Thus, in the first instance, the interaction of the Zn (II) ion with fluorescein, containing five active oxygen atoms, producing two types of Zn (II)–fluorescein complexes, was investigated. Since, the energy of the second complex, Zn (II)–fluorescein-(2), was lower, its interaction with AFB_1_ was investigated. According to the obtained data, zinc ions preferentially bind to both oxygen atoms in the fluorescein molecule; moreover, the Zn–fluorescein complex interacts with AFB_1_; see Figure 13, a four-coordination number is observed. As a complementary result, it is important to note that the corresponding dipole moment values for the obtained complexes support the stability of the ternary complexes. To highlight the reactive regions of aflatoxin and fluorescein molecules, their molecular electrostatic potential was appropriately measured. Moreover, the structure of the Zn (II)–fluorescein–AFB_1_ complex confirmed both its stability and molecular properties for possible future applications.

(v). A quantitative description of the chemical interactions among one and two AFB_1_ molecules with chlorophyll a (chl a) was obtained by Vázquez-Durán et al. [33], using the M06-2X functional with the 6-311G(d,p) basis set, considering the gas and water phases. In this sense, molecular properties such as the molecular electrostatic potential surface, HOMO and LUMO, and charges were employed to characterize and describe the interaction sites. The molecular electrostatic potential map showed the chemical activity for various sites of the AFB_1_ and chl a. The energy difference between the molecular orbitals of AFB_1_ and chl a allowed the establishment of an intermolecular interaction. Moreover, chl a showed two conformations, unfolded and folded, with a difference of 15.41 kcal/mol. In the computed ground state of chl a, the Mg^2+^ ion could be axially coordinated to β (up) and α (down) by the oxygen atom of the ketone group, ring E, or by the oxygen atom of the carbonylic lactone group, ring D, C10=O21 or C8=O20, respectively. On the other hand, molecular dynamics (MD) simulations of chl a folded showed conformational changes, meaning that it probably exists both folding and unfolding in equilibrium. When the interaction between one AFB_1_ molecule and the chlorophyll was β-oriented, the corresponding coordination product appeared more stable, with −39.2 kcal/mol. However, when the interaction was between two AFB_1_ molecules (ring E) and chl a folded, it was the most stable, with −64.8 kcal/mol. The acquired energy interactions between chl a unfolded, chl a folded, and AFB_1_, considering water as the solvent, were lower than those shown in the gas phase. This is probably because chl a unfolded and chl a folded undergo a more significant interaction with the medium than with AFB_1_. These findings were supported by the results achieved in the docking studies, with the interaction between chl a folded and one AFB_1_ molecule being the most stable, because the folded conformation is preferred. Finally, the complexes with two AFB_1_ molecules were more stable than those with only one AFB_1_. Thus, it is essential to highlight that biosorbents containing chlorophyll could improve AFB_1_ adsorption.

### 3.4. Molecular Interactions with Environmentall Compounds (Clays)

Computational chemistry methodologies have also been used to simulate and to interpret the adsorption mechanism of the AFB_1_ metabolite in three clays: smectite, illite and kaolinite, to analyze its elimination, to reduce its presence, and its toxicity using eco-friendly materials; consequently, it was considered convenient to include the following summaries.

(i). In an interesting work, Deng and Szczerba evaluated computationally: (1) the bonding mechanism between AFB_1_ and smectite, and (2) the adsorption models of the AFB_1_–smectite complexes [53]. The obtained results indicated that more than 96% of interactions between exchange cations (Na^+^ and Mn^2+^) and the AFB_1_ molecule occurred with the two carbonyl oxygens (O20 and O21, Figure 1) in the interlayer of the smectite. Non-significant interactions were observed with the lactone ring oxygen (O7) and the dihydrofuran oxygens (O1 and O18). The surface electrostatic distribution study confirmed that these two carbonyl oxygens were the most significant reaction sites when AFB_1_ was coordinated with the positively charged exchange cations in the smectite. However, the dihydrofuran oxygens in the AFB_1_ molecule were the subsequent possible reacting sites. Dynamics simulation (Figure 14) showed that molecules interacted by docking one exchange cation of the smectite into two carbonyl oxygens of the AFB_1_ molecule; however, individual interaction between one of the two carbonyl oxygens with one cation was also observable. In general, after interacting AFB_1_ with the different exchange cations via ion–dipole interaction or coordination (under fully and partially dehydrated conditions), cations possessed a less positive charge, meaning that electrons shifted from AFB_1_ toward the exchange cations. Finally, when cations (K^+^, Na^+^, Ca^2+^, Mg^2+^, and Mn^2+^) interacted with AFB_1_, computed vibrational studies indicated that several AFB_1_ infrared bands shifted, and the band intensity changed considerably. Thus, Deng and Szczerba pointed out the importance of carbonyl groups in bonding AFB_1_ to smectite.

(ii). In an interesting paper, Kang et al. in 2016 [54], also discriminated computationally the bonding types between AFB_1_ and clays, such as kaolinite, illite, and smectite, in addition it was also examined the association of the physical bonding strength/energy with AFB_1_. In general, the gradient isosurface study suggested that AFB_1_ was bonded to kaolinite via weak hydrogen bonding by a pair of weak bonds [(Si/Al-OH)_2_· · · (O=C)_2_]. Furthermore, AFB_1_ and illite interacted through a K^+^ bridge (C=O· · ·K^+^· · ·O-Al). This interaction was associated with the coordination of K^+^ to electronegative oxygen atoms in AFB_1_ and illite. These results indicate that AFB_1_–illite and AFB_1_–kaolinite complexes are associated with electrostatic interactions such as weak hydrogen bonding and moderate electron–donor–acceptor attraction, respectively. Furthermore, AFB_1_ was bonded to smectite by the release of Ca^2+^ from the interlayer space of the clay. Thus, Kang et al. proposed that Ca^2+^ was synchronously bonded to two carbonyls of the AFB_1_ molecule. Nevertheless, Ca^2+^ was also attracted to the four oxygen atoms of the Si-O ring on the surface of the smectite. Consequently, a strong calcium-bridging linkage [(C=O)_2_· · ·Ca^2+^· · ·(O–Si)_4_] was involved in the AFB_1_–smectite adsorption. In addition, the ΔG^0^-based ln*K_d_* was calculated at the DFT B3LYP/6-31G(d,p) level. Results showed that the adsorption of AFB_1_ to the three clays appeared in the order: AFB_1_–smectite > AFB_1_–illite > AFB_1_–kaolinite, suggesting that the bond energies increased in the same order. Finally, based on the ΔG^0^-based ln*K_d_* results, the authors concluded that the sorption of AFB_1_ to clays involves a spontaneous physisorption mechanism because all ΔG^0^ values were lower than 20 kJ/mol.

### 3.5. Molecular Interactions with Biological Compounds (DNA, Enzymes, Cyclodextrins, Glucans, among Others)

The last section was accomplished taking in mind that, molecular simulation methods arose as a necessity in different areas of science with the aim of studying those molecules that can’t be observed at the experimental level, that present equilibrium times on such small scales (of the order of nanoseconds) or whose molecular dynamics (MD) can be better understood using computational tools. Among some of the applications in which these methods have been of great help, we can mention the study of macromolecules such as DNA. Although this methodology requires validation, it helps predict or corroborate experimental results, thus reducing the number of resources needed for experimentation. Since MD allows to know if an interaction is stable over time; hence, for a ligand to act in a specific place, it is necessary to know if the affinity (interaction energy) is feasible, as well as if the interaction with a biomolecule is possible explaining the toxicity, excretion, and biological action, among other activities.

According to the aforementioned commentaries and regarding the methodology in Section 3.1, the determination of some molecular properties has allowed the establishment of the several interaction types (hydrogen bond, hydrophobic, dispersion, pi-pi, and pi-alkyls); additionally, to union forces to explain the coupling of AFB_1_ and biomacromolecules. Computational chemistry methodologies also allow us to understand the chemical reactions between AFB_1_ and biomacromolecules to determine if the complex is going to be formed, reproduce any experimental data or displays affinity for some type of protein. Moreover, at molecular level, it is possible to analyze what type of binding could exist between AFB_1_ and the substrate (albumin) and this could favor the transport of molecules. finally, with these methodologies is possible to obtain the ΔG, indicating if the interaction is spontaneous and favorable.

(i). To clarify the interaction between AFB_1_ and DNA, Loechler et al. in 1988 [55], applied molecular modeling employing the PSFRODO program and combining molecular mechanical calculations with the AMBER program. Loechler et al. confirmed that AFB_1_ accomplishes an adduct exclusively at guanine’s N(7) position in different DNA sequences, concluding that two approaches seem reasonable: (a) the AFB_1_ moiety is intercalated between the base pair containing the adducted guanine and the adjacent base pair on the 5′-side regarding the adducted guanine, and (b) AFB_1_ is bound on the exterior of the major channel of DNA; therefore, due to the rotational flexibility of AFB_1_ during the interaction with DNA sequences, four potential binding sites resulted in agreement with the “clock”.

In addition, it was specified that, ignoring the high energy structures of DNA; only two binding modes seemed feasible for AFB_1_ both bonded at the N(7) position of guanine in DNA; these depended upon the conformation of the terminal, saturated furane ring on the aflatoxin moiety bonded to DNA: with a boat-like conformation of the oxolane structure, the aflatoxin acted in an intercalative binding manner in DNA, whereas with a chair-like conformation, aflatoxin lay externally in the major groove of the DNA. The former was the conformation observed in the crystal structure for AFB_1_ [7,8]. Finally, a transition state model related to the reaction of AFB_1_ with the N(7) position of guanine in DNA is also suitably suggested.

(ii). Bonnett and Taylor employed two theoretical models –intercalated (noncovalent) and covalent—to form the complex between AFB_1_ and the amino acid guanine of DNA, employing classical empirical molecular energy potential functions [56]. Bonnet and Taylor proposed that the covalent binding of AFB_1_–DNA was preceded by an intercalative interaction and various conformational changes were perceived, due to the orientation of the AFB_1_ molecule with respect to the surface of DNA; hence, the process occurs by two associations, 3′ and 5′, of AFB_1_ with the target of the nucleoside. On the other hand, the covalent process shows the same associations, although it is important to highlight that these interactions occur with the BP2 and BP3 Gua bases. In this sense, the covalent bond between AFB_1_ and BP2 Gua was with the 5′-side nucleosides, and for BP3, it is at the 3′-side of the nucleoside. In addition, in a similar pathway to the AFB_1_ epoxide, which suffers from the formation of carbonium ions and then an electrophilic attack, the AFB_1_ complies with this general process, leading to a covalent bond with DNA (N7-Gua), and consequently, the intercalation process is avoided.

(iii). Yiannikouris et al. in 2006 [57], conducted several molecular modeling studies to investigate the accessibility of AFB_1_ inside the polysaccharide chains of β-D-glucans (one of the major constituents of the cell wall of *Saccharomyces cerevisiae*). In silico molecular mechanics investigations were applied in an attempt to elucidate the nature of the interaction and to identify the functional groups involved in the docking process. Docking of the AFB_1_ molecule inside the β-D-glucan structure includes a two-step mechanism: (1) AFB_1_ is trapped inside the single helix of the (1→3)-β-D-glucan chain, and (2) the branched (1→6)-β-D-glucan chain covers the AFB_1_ molecule and maintains it inside the helix. Based on the results of the AFB_1_ interaction with β-D-glucans, Yiannikouris et al. concluded that the (1→3)-β-D-glucan single helix was of primary importance in establishing the interaction with AFB_1_, whereas (1→6)-β-D-glucans strengthen the van der Waals interactions. In general, hydrogen bonding involving the hydroxyl, lactone, and ketone groups of the AFB_1_ molecule that occurred during the interaction with β-D-glucans. However, an interaction due to van der Waals forces played a significant role in the binding process of AFB_1_ to β-D-glucans, accounting for 93.4% of the total docking energy.

(iv). A model of guanine (Gua) alkylation by AFB_1_ *exo*-2,3-epoxide was studied by Bren et al. [58]. It is important to highlight, that this reaction represents an initial step of carcinogenesis associated with AFB_1_. For this purpose, the DFT/B3LYP and HF levels of theory, in conjunction with the 6-31G(d), 6-31+G(d,p), and 6-311++G(d,p) basis sets, were employed, in addition to the use of semiempirical MO methods AM1 and PM3 (parametric method 3). Complementarily, two methods to calculate solvation-free energies were used: the solvent reaction field (SCRF) and the Langevin dipoles model (LD) parametrized. The SCRF method was applied at all ab initio and DFT levels. The obtained Merz–Kollman partial atomic charges served as the input for the LD model. In silico-calculated activation free energy resulted in agreement with the experimental value of 15.1 kcal/mol. An interesting additional result corresponded to the free energy preference of 0.49 kcal/mol for AFB_1_ *exo*-2,3-epoxide over the *endo* stereoisomer for the response with Gua, in conjunction with steric hindrance; see Figure 15. Finally, the authors conclude that the stereoselectivity of this reaction exists in the aqueous solution and is further enhanced in the DNA duplex. Furthermore, calculations at all theory levels predict a lower activation free energy for the similar reaction between 3a,6a-dihydrofuro[2,3-*b*] furan (DHFF) *exo*-4,5-epoxide (a truncated version of AFB_1_ *exo*-2,3-epoxide) and guanine.

(v). The importance of this search was appointed improving the fluoresce of AFB_1_ in solution, a spectrofluorescence technique was performed allowing to Aghamohammadi and Alizadeh to understand the interaction between AFB_1_ and three different cyclodextrins (CD) (α, β, and heptakis-2,6-dimethyl-*o*-β-CD (ome)), employing different concentrations in aqueous solutions; see Figure 16 [59]. This last inclusion complex (ome) depicted the greatest fluorescence intensity, due to the incorporation of AFB_1_ in the non-polar cavity, being attributed to the specific complex formed between CD and AFB_1_. It is important to note that the geometry of the complexes was investigated by a molecular modeling approach employing the semiempirical HF–SCF calculations, finding in the first instance the most favorable structure. In addition, the hydrophobic interactions between the host and guest appeared weak, or the cavity of CD was not filled with AFB_1_ molecules. In this sense, the AFB_1_ and molecules β-CD were optimized by molecular quantum mechanics studies employing the AM1 method and B3LYP/631++G** level. This study provided evidence that the van der Waals interactions are responsible for the 1:1 complex formation stability. Consequently, due to good features such as the internal cavity diameter and depth of β-CD (6.2 and 7.8 Å, respectively), the AFB_1_ molecule has partial inclusion in the β-CD; see Figure 17a. In this sense, the furan ring of the AFB_1_ molecule is placed inside the cavity of β-CD, Figure 17b, with respect to the cyclopentanone ring as reference’s plane.

(vi). Ramírez-Galicia et al., in 2007 [60], reported a theoretical study, related to the fluorescent enhancement of the AFB_1_–β-cyclodextrin complex in the presence of water. For the corresponding purpose, in the first instance, the AM1 method and further DFT using the B3LYP/6-31G* level were employed to optimize the AFB_1_ molecules and their complexes with β-CD and water. Thus, the authors reported that the fluorescence of AFB_1_ is due to two configurations with the electronic transfer of one electron from the corresponding HOMO to LUMO. When the water molecules are located near the carbonyl group of the AFB_1_ molecule, a vibrational coupling between the bending modes of the water molecules, by hydrogen bond, with the carbonyl group exists, and the net result would be fluorescence quenching, since the AFB_1_ molecules involve an electronic excitation in this coupling process. However, when β-CD is added to the solution, the AFB_1_ molecule forms a complete set of interactions with the surrounding water molecules; thus, the fluorescence is improved by the lesser interaction of the water molecules with the carbonyl group of AFB_1_.

(vii). An interesting search was performed by Bren et al. asserting that the 3A4 isoform of CYP450 (CYP3A4) displays positive homotropic cooperativity towards AFB_1_, which is produced by the simultaneous occupancy of the large active site by multiple AFB_1_ molecules [61]. In this sense, a series of computational studies were performed. For this purpose, CYP3A4–AFB_1_ complexes, including one and two molecules of AFB_1_, were created, performing docking calculations. In this sense, the obtained docking studies engendered three plausible doubly ligated CYP3A4 complexes with AFB_1_, which MD simulations later analyzed. Bren at al. showed that one AFB_1_ molecule was bound in a face-on C2−C3 epoxidation, and a second AFB_1_ molecule was bound in a side-on 12α-hydroxylation mode (Figure 18). The empirical linear interaction energy method revealed that shape complementarity through non-polar dispersion interactions between the two bound AFB_1_ molecules was the main source of the experimentally observed positive homotropic cooperativity.

(viii). Dellafiora et al., in 2017 [1], modeled the interaction of AFB_1_ with three laccase enzymes (beta, delta, and gamma isoforms) from the white rot fungi *Trametes versicolor*. Dellafiora et al. reported that the docking simulation of AFB_1_ satisfactorily interacted with both beta and delta laccase isoforms. Hydrophobic–hydrophobic interactions mainly drove these interactions. However, a single polar interaction was also present, where the oxygen of the methoxy group (O22, Figure 1) of the AFB_1_ molecule engaged the His458 with a hydrogen bond. Conversely, AFB_1_ did not interact with gamma isoforms. This phenomenon was mainly due to the incapability of AFB_1_ to sink within the enzyme’s catalytic site. It has been reported that multi–copper–containing enzymes such as laccases can oxidize a broad range of substrates, including AFB_1_. However, attention should be paid to these, since the significant structural differences between isoforms limit the accessibility of the substrate to the catalytic site. Moreover, the precise degradation mechanism and the structure of the degraded products have not yet been established.

(ix). Related to the studied protein, to determine the binding bonds between these and aflatoxins (AFB_1_, AFB_2_, AFM_1_, AFG_1_, and AFG_2_), Wang et al. in 2017 [62] conducted a combination of computational studies comprising target fishing, docking studies, MD simulations, and MM/PBSA calculation. Hence, twenty proteins were judged, considering compounds sharing structural similarities with similar target association profiles. After analyzing the binding properties of protein–ligand complexes obtained by docking studies, Wang et al. selected three complexes, which were submitted to 2 ns of MD simulations, employing the Amber12 program. In addition, examining the binding free energies, it was possible to corroborate the stability of the complexes. According to the results, Wang et al. proposed that trihydroxynaphthalene reductase, Glycogen Synthase Kinase 3 Beta (GSK-3b), and Moloney Murine Leukemia virus 1 (Pim-1) could be the target proteins for AFB_1_. It is important to note, that the binding of all the protein–AFB_1_ complexes was dominated by van der Waals interactions. This study, also allowed to Wang et al. to point out that GSK-3b was the most potent AFB_1_-binding protein.

(x). de Almeida et al., related to acetylcholinesterase (AChE) inhibition, employed computational studies to evaluate, the binding mode of AFB_1_ and related aflatoxins to the catalytic anionic site (CAS) of AChE [63]. In this regard, the docking studies showed that all the studied compounds interacted with the CAS exhibiting negative ΔG values, while AFP_1_ and AFQ_1_ showed the highest affinity. Related to the protein–ligand complexes obtained by docking studies, they were submitted to 50 ns MD simulations to calculate the free energy of the total system; in this case, the obtained results showed that all systems reached confluence, probably due to AFB_1_ and its metabolites interacting with the CAS site on AChE near Asp74 and Trp86, coming approaching to its active site. Finally, MM-PBSA studies corroborated the affinity of AFB_1_ and its metabolites to the peripheral anionic site (PAS) of AChE. The integration of both studies allowed to conclude that AFB_1_ shows two binding modes: on PAS, acting as a non-competitive inhibitor, and on CAS, close to the active site, acting as a competitive inhibitor.

(xi). Experimental and computational studies, to interpret at an atomic level AFB_1_–human serum albumin (HAS) interactions, were performed by Bagheri and Hossein Fatemin in 2018 [64], employing fluorescence spectroscopy, docking studies and MD simulations. Bagheri and Hossein Fatemin found that the binding mode of AFB_1_ with HAS was located at site I of subdomain IB—see Figure 19—by hydrophobic interactions (Ala158, Tyr161, Leu139, Leu 135, Ile142, Tyr138, Phe134, Phe165, Leu182, and Met123), and a hydrogen bond with amino acid Arg117, resulting in good agreement with experimental fluorescence data. Hence, HAS fluorescence emission was quenched with the addition of AFB_1_ at different concentrations; the obtained result is indicative of complex formation. Furthermore, employing thermodynamic parameters, it was possible to propose that serum albumin–AFB_1_ recognition is spontaneous, where the interactions by hydrogen bonds and van der Waals are the main forces.

(xii). Related to the previous work by de Almeida et al. [vide supra], a set of molecular modeling studies were performed determining the binding mode of AFB_1_ and its metabolites (aflatoxicol (AFL), 2,3-epoxide (AFB epoxide), AFQ_1_, AFP_1_, AFB_2a_, and AFM_1_) inside the PAS of AChE [65]. Thus, in the first instance, docking calculations employing Molegro Virtual Docker were performed, for the studied compounds, showing that AFM_1_ and AFB epoxide showed the highest affinity to AChE, by interacting with amino acid residues Tyr72, Tyr124, and Val294, located in the PAS (Figure 20). Then, the protein–ligand complexes were submitted to 50 ns of MD simulations, showing that the total energy tends to reach stability after 2 ns of simulation for all the complexes. In addition, by MM-PBSA calculations, it was possible to corroborate that these aflatoxins metabolites exhibited lower energy values than AFB_1_, reinforcing the idea that AFB_1_ metabolites could be better AChE inhibitors.

(xiii). Regarding the protein effects, the potential binding of GSK-3b with AFB_1_ was reinforced by Hailu et al. in 2019 [66], employing the PatchDock server, in addition to molecular physicochemical, drug-likeness, and ADME (Absorption, Distribution, Metabolism and Excretion) analyses of AFB_1_ and another twelve analogs. It is convenient to note that, all the studied compounds aligned well with Lipinski’s rule of five, and the ADMET profile predicted a high gastrointestinal absorption effect for all molecules. Moreover, according to the docking results, the AFB_1_ displayed a subtle atom contact energy (−168.64 kcal/mol), but it was highlights that AFB_1_ does not exhibit an interaction with GSK-3b amino acid residues.

(xiv). Chen et al. 2019 [67] aimed to discover protein targets to explain the effects of AFB_1_, AFB_2_, AFM_1_, AFM_2_, AFG_1_, and AFG_2_; consequently, they performed some docking studies, employing Discovery Studio 3.1 software. The obtained results showed that AFB_1_ displays high affinity to estrogen sulfotransferase. AFB_1_ binds to the cavity of estrogen sulfotransferase to establish a cation–π interaction between the benzene ring and Arg129 residue, while the carbonyl group in coumarin forms hydrophobic interactions with the Tyr192 residue and the oxygen atom on the furan ring forming hydrophobic interactions with the Arg256 residue. The proteins included in the study play significant roles in cell apoptosis, estrogen metabolism, immunosuppression, and digestive system function.

(xv). Zhang et al. [68] developed a sensitive immunoassay to detect AFB_1_, through Fab (antigen-binding fragment). At the experimental level, Fab showed a Kd value of 1.09 × 10^−7^ mol/L to AFB_1_. Assessment of the interactions between the antibody and AFB_1_, molecular docking, molecular dynamic simulation, and quantum chemical evaluations were performed on AFB_1_ in conjunction with a model of the Fab fragment. In this sense, the electrostatic potential (ESP) analysis of AFB_1_ at the B3LYP/6-311++G(d,p) level was performed to explore the electrostatic interactions (recognition process) between AFB_1_ and Fab. As can be seen in Figure 21, the most negative ESP region (dark red color) with a global minimum of −60.88 kcal/mol corresponds to the ketonic oxygen atoms. Due to a lone pair of electrons of the oxygen, this molecular segment causes interactions with the Trp98 residue, contributing −6.96 kcal/mol of the electrostatic interaction energy. Moreover, the Ser32 and Trp98 residues, were nearest to the AFB_1′_s minimum and maximum global ESP points, contributing almost 46.8% of the electrostatic interaction energy in the AFB_1_–Fab joining process. Finally, Zhang et al. concluded that the hydrogen bonds and π-π stacked/π-alkyl interactions represent the key interactions for the antibodies’ recognition in AFB_1_; in addition, van der Waals interactions played more important roles than electrostatic interactions in maintaining the stable Fab–AFB_1_ combination, where the Ser32, Trp93 and Trp98 residues played the main role. Finally, the carbonyl and methyl groups in AFB_1_ were considered the strategic chemical structures recognized by the Fab antibody.

(xvi). Mousivand et al. created a new aptamer of DNA, over AFB_1_ for a biosensor development by means of a genetic algorithm (GA) based on an in silico maturation strategy [69]. High affinity and selectivity were attained by modifying the sequence of a known AFB_1_ aptamer, previously registered in a patent [70]; in this sense, Mousivand et al. employed seven GA rounds and three generations of single-stranded DNA (ssDNA) oligonucleotides, considering them, new candidate aptamers.

Additionally, a new pipeline was applied, to know the tertiary structure of all single-stranded DNA sequences. As can be seen in the corresponding paper [69], the molecular docking methodology was used to investigate the best AFB_1_ aptamer, and to evaluate their corresponding binding affinity and selectivity toward AFB_1_, the aptamers F20, g12, C52, C32, and H1 were applied as probes in an unmodified gold nanoparticle. Subsequently, the molecular docking study showed that the best affinity corresponded to the F20 aptamer, located in the loop and stem regions, by the interaction of a hydrogen bond and a hydrophobic interaction over the AFB_1_ molecule; see Figure 22. In addition, using a colorimetric aptamer assay (fluorescent anisotropy), considering gold nanoparticles, it was conveniently recognized as the best aptamer–AFB_1_ complex; it also was confirmed that the F20 aptamer carries the highest affinity toward AFB_1_, revealing 70% of agreement between the experimental and the in silico studies. Finally, it is necessary to point out that the F20 aptamer similarly displayed the highest selectivity to AFB_1_.

(xvii). In an in silico pipeline study, ten single-stranded DNA (ssDNA) aptamers against AFB_1_, fumosine B1, and ochratoxin A were investigated [71], determining the binding affinity. For this work, the AFB_1_ aptamers (AF_AB3 and AF_APT1) were considered, comparing both experimental studies of fluorescent microscale thermophoresis and magnetic bead assays. For this purpose, several free computational software programs—SimRNA, RxDOck, BioPython, and Open Babel—were employed. Thus, the most stable aptamers for AFB_1_ were AF_AB3 and AF_APT1, with binding free energy ΔG^0^ of −9.20 and −10.66 kcal/mol, respectively. Related to the comparison with the fluorescent microscale thermophoresis assay, the AF_AB3 aptamer showed good binding toward AFB_1_, with constant dissociation (*K_D_*) value of 178 nM and ΔG^0^ of −9.21 kcal/mol; however, the AF_APT1 aptamer did not display any binding toward AFB_1_. Related with the magnetic bead assays, the AF_APT1 aptamer displayed the best binding to AFB_1_, with *K_D_* value of 1.40 nM and ΔG^0^ of −12.08 kcal/mol, and the AF_AB3 aptamer did not display any binding toward AFB_1_.

(xviii). In a relative recent research, Liu et al. in 2020 [72] conducted MD simulations to predict the interaction between a recombinant *Trametes* sp. C30 laccase from *Saccharomyces cerevisiae* and AFB_1_. The computational study pointed out that AFB_1_ satisfactorily interacted with laccase and the oxygen atoms of the methoxy group, and the terminal furan ring of the AFB_1_ molecule formed hydrogen bonds with the amino acids His481 and Asn288, respectively. MD simulations studies indicated that the residue His481, present in the coordination sphere of the T1 copper, interacts with the AFB_1_ molecule, thus mediating the electron transfer during oxidation. Liu et al. concluded that the C30 laccase and the AFB_1_ molecule interacted mainly via hydrogen bonds.

(xix). In the last considered search, Aamir Qureshi and Javed, in 2020 [73], studied the interaction between chicken serum albumin (CSA) and AFB_1_, using fluorescence spectroscopy, UV–Vis spectroscopy, and molecular docking. The emission fluorescence from CSA displayed quenching promoted by increasing the concentration of ligand AFB_1_. In the case of the UV–Vis results, a hyperchromic effect for the complex in the ground state at different concentrations was revealed, displaying a significant blue shift. Related to the thermodynamic parameters, the negative ΔG value was characteristic of spontaneous and favorable interactions between CSA and AFB_1_, being the key interactions the hydrogen bond and van der Waals forces. MD simulations allowed the corroboration of two hydrogen bonds between AFB_1_ with amino acid residue Arg139 from CSA, and hydrophobic interactions with amino acid residues Pro106, Tyr110, Asn107, Lys105, Phe112, Ile103, Ser111, and Thr104, permitting the recognition of AFB_1_ by CSA.

Finally, Table 3 summarizes all the outstanding aspects, all of them separated by the previously considered sections in this work, taking in mind the chronological order.

## 4. Conclusions

In this paper, after a profound literature search and then appropriately organizing the obtained information, it was achievable to engender the first review related to the theoretical studies (quantum chemistry–computational chemistry) of aflatoxin B_1_. It is important to note that AFB_1_ has the most potent mutagenic and carcinogenic activity among the known aflatoxins. However, it was found that some AFB_1_ derivatives (epoxidation, demethylation, acid hydrolysis, and carbonylic reduction) enhance or diminish AFB_1_’s toxicity. It is also important to highlight that the presented information aimed to provide an understanding of how AFB_1_ and its derivatives can bind to DNA or proteins to cause damage and inhibit acetylcholinesterase or human serum albumin. It was also possible to present strategies to quantify and detect AFB_1_ in different systems and to be able to eliminate or deactivate AFB_1_ by trapping or modifying its structure, emphasizing that some natural strategies (employment of clays and biosorbents) yield excellent results.

Furthermore, taking into account the important and interesting information assigned in the manuscript of this review, it is conveniently feasible to explain that the theoretical predictions of the structure and properties of AFB_1_ have advanced significantly since the twentieth century. In this sense, computational methods such as semiempirical methods, Hartree–Fock, DFT, molecular docking, molecular dynamics, machine learning, and artificial neuronal networks have been conveniently applied.

It is also important to highlight, bearing in mind future perspectives, that with all the selections appropriately summarized, the stakeholders in this area could conveniently and suitable expand their research toward another toxic derivative, such as the AFB_1_ epoxide, to propose strategies to diminish their toxicity and to identify their structure–activity relationships. In addition, different AFB_1_ derivatives generated by the metabolism from livestock, for example AFM_1_ and AFL, could be studied. Additionally, the studies could be lengthened in order to know the adsorption behavior of AFB_1_ and its derivatives using other clays. In addition, by docking and the molecular dynamic, the interactions between AFB_1_, its metabolites, and some biomacromolecules could also be studied. Consequently, at present, unequivocally theoretical studies of the AFB_1_-derivatives may be a promising research field.

On the other hand, it is important to mention some limitations of the methods. In this case, for quantum chemistry-computational chemistry, the results depend on the method employed: quantum mechanics, molecular mechanics and bioinformatics; additionally, to the level of theory: semi-empirical, Hartree–Fock, and density functional theory, since depending of the chemical system, the computational requirements could be greater. 

## Figures and Tables

**Figure 1 toxins-15-00135-f001:**
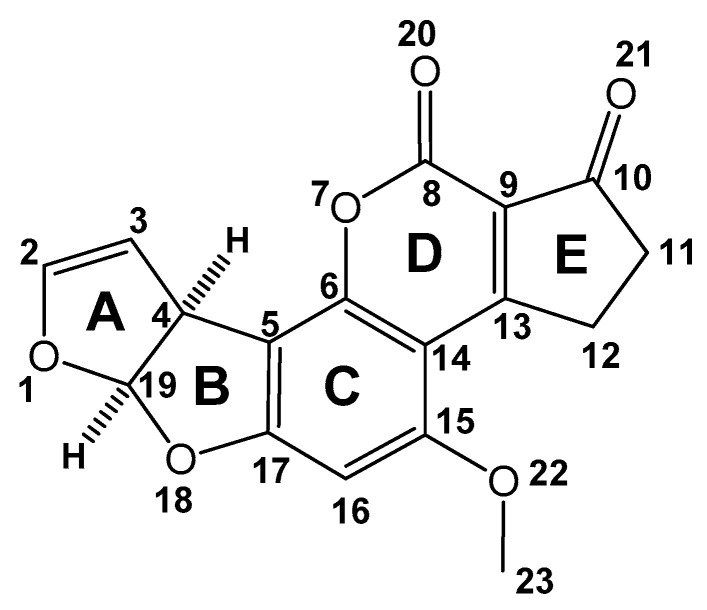
Structure of AFB_1_. The numbers are indicative of propose of numeration for AFB1, and the capital letters are label to identify the different rings in AFB_1_.

**Figure 2 toxins-15-00135-f002:**
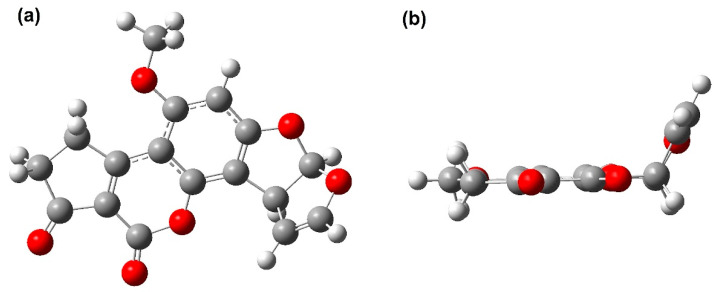
(**a**) Optimized and (**b**) planar structure of AFB_1_. Oxygen is identified by red ball, carbon by grey ball, and hydrogen by white ball.

**Figure 3 toxins-15-00135-f003:**
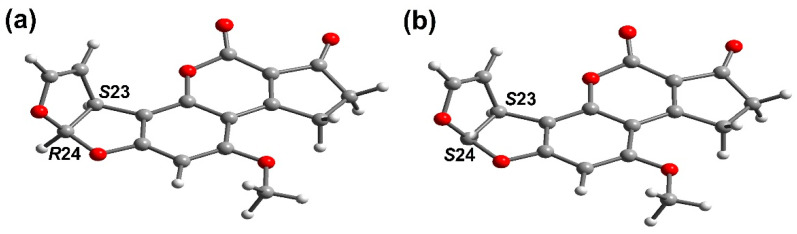
Optimized geometric structure of (**a**) *cis* AFB_1_, and (**b**) *trans* AFB_1_. Adapted from Ref. [39].

**Figure 4 toxins-15-00135-f004:**
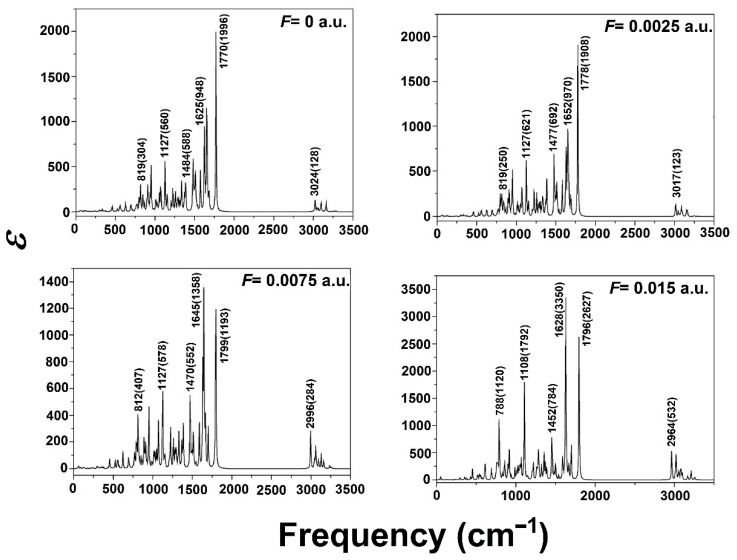
Some IR spectra of AFB_1_ in external fields. Adapted from Ref. [40].

**Figure 5 toxins-15-00135-f005:**
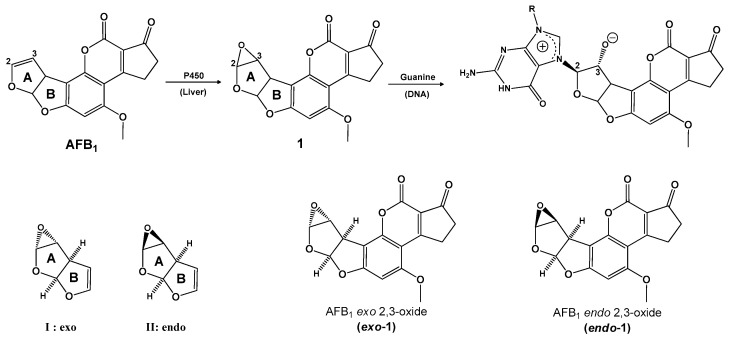
Ring opening of AFB_1_ epoxide (**1**), compound models **I** and **II**, and AFB_1_ epoxide adducts. Adapted from Ref. [41].

**Figure 6 toxins-15-00135-f006:**
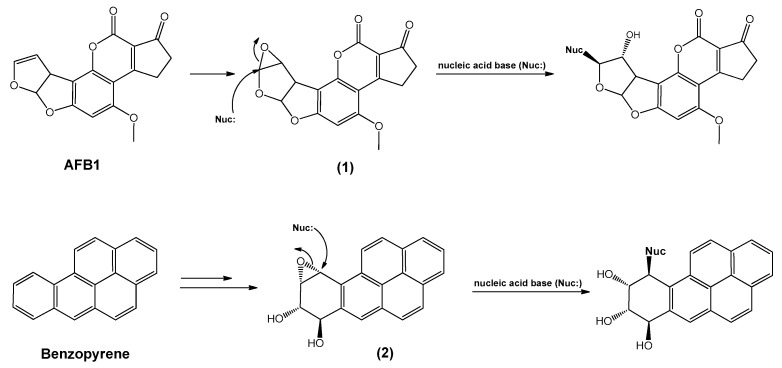
Intercalating complexes and ligand–DNA adducts of the mutagenic aflatoxin β1-2,3-oxide (**1**) and benzo[*a*]-pyrenediol oxide (**2**) and their reactivity for a SN_2_ type oxirane ring opening for (**2**). Adapted from Ref. [42].

**Figure 7 toxins-15-00135-f007:**
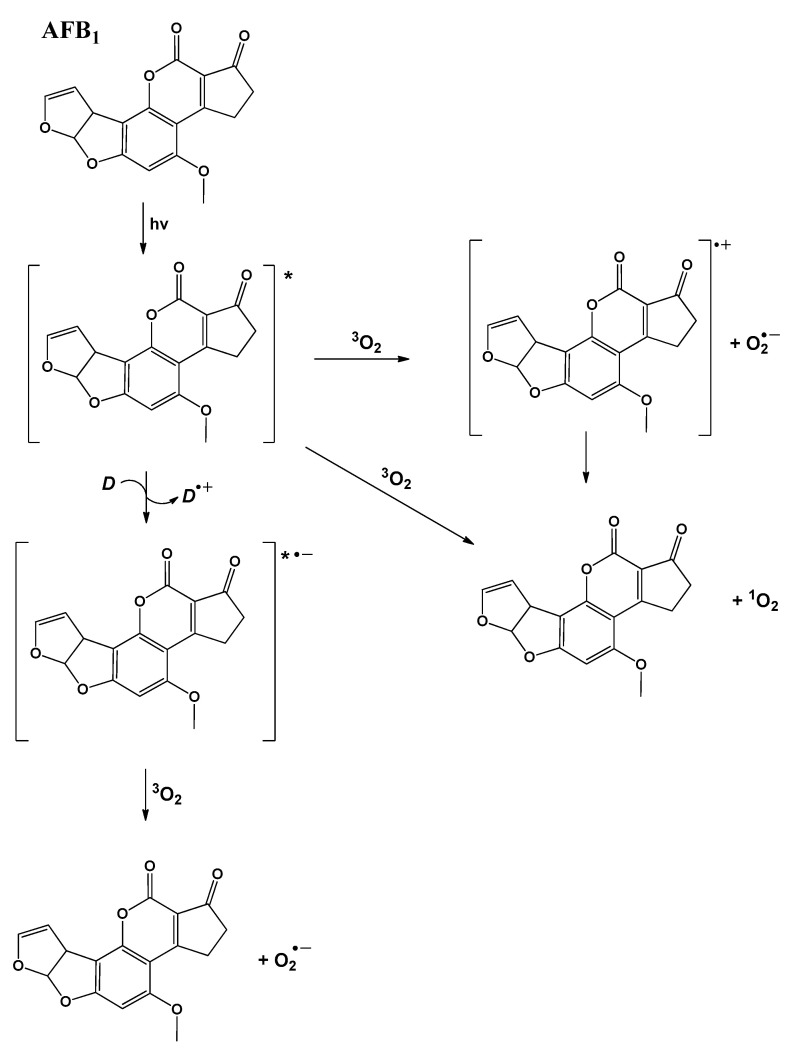
Phototoxicity pathways of AFB_1_. Adapted from Ref. [43]. *: excited state.

**Figure 8 toxins-15-00135-f008:**
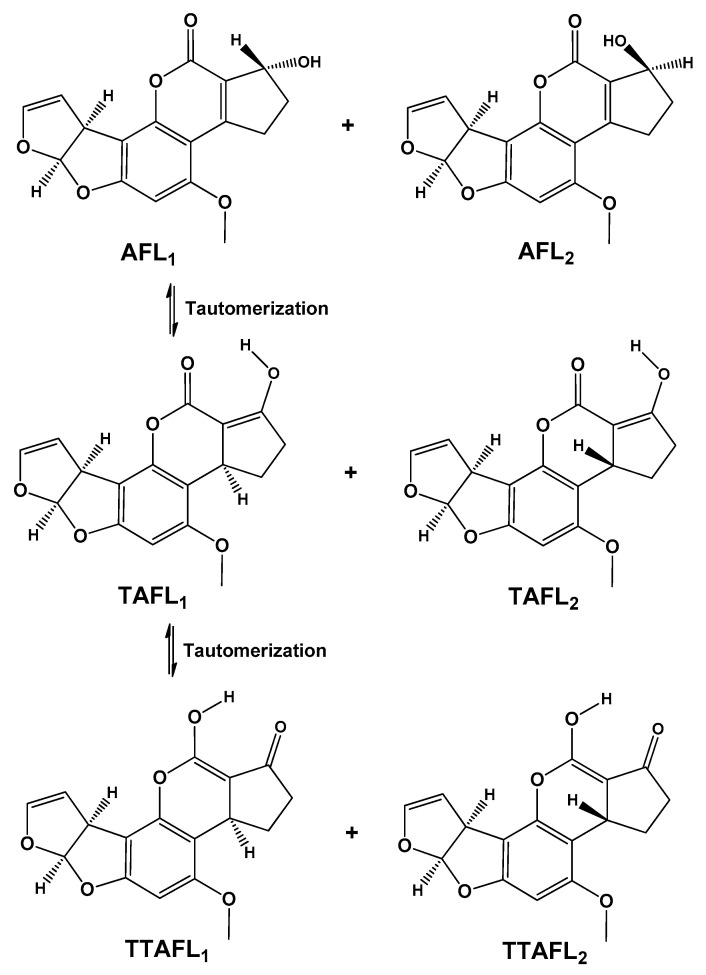
Stereoisomers and tautomers of AFL. Adapted from Ref. [45].

**Figure 9 toxins-15-00135-f009:**
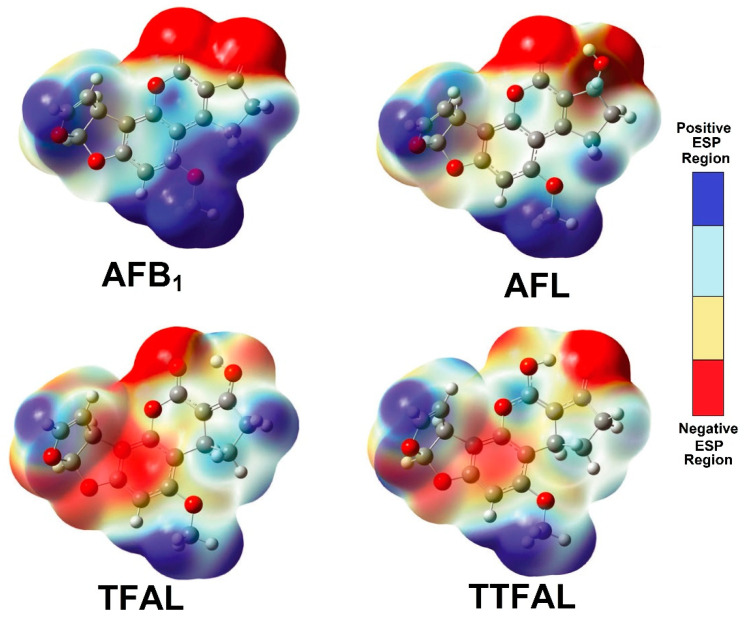
Electrostatic potential energy maps, isovalue = 0.02. Adapted from Ref. [45].

**Figure 10 toxins-15-00135-f010:**
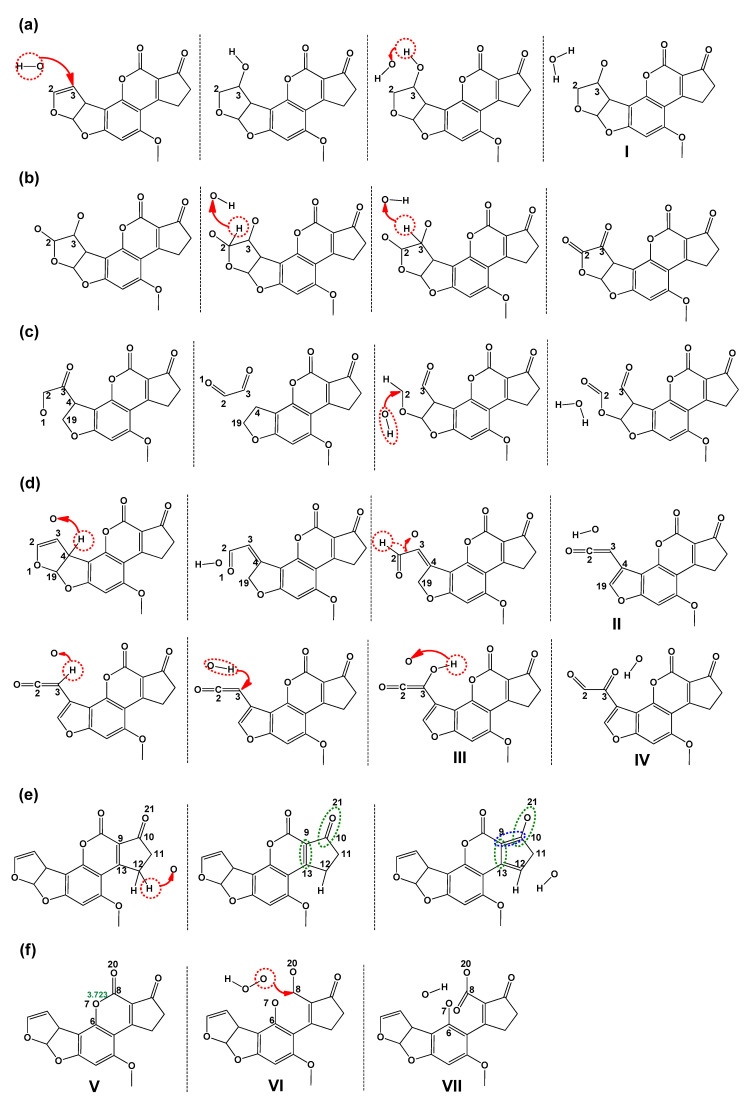
MD simulation of (**a**) addition of OH radical, (**b**) oxidation of C2-C3 bond, (**c**) rupture of ring A by dissociation of C19-O1 and C3-C4, and by rupture of C2-C3 bond (**d**) dissociation of ring A by C19-O1 and dialdehyde formation, (**e**) dissociation of ring D by reduction of C9-C13 and carbonyl bond C10-O21, and (**f**) cleavage by elongation and breakage of O7-C8 and C8-C20. The red circle is related to ROS molecules or the hydrogen subtraction; the red arrow indicates the position of addition; green circle indicates the reduction of bonds; blue circle indicates the formation of new double bond; the green numbers show the distance bond, and black numbers are related to position, according to Figure 1. Adapted from Ref. [47].

**Figure 11 toxins-15-00135-f011:**
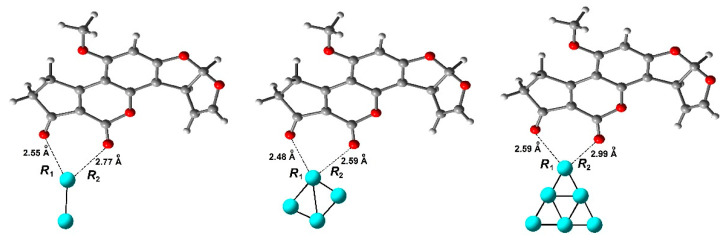
Optimized structures of AFB_1_-Ag (n = 2, 4, 6) complexes. Adapted from Ref. [50].

**Figure 12 toxins-15-00135-f012:**
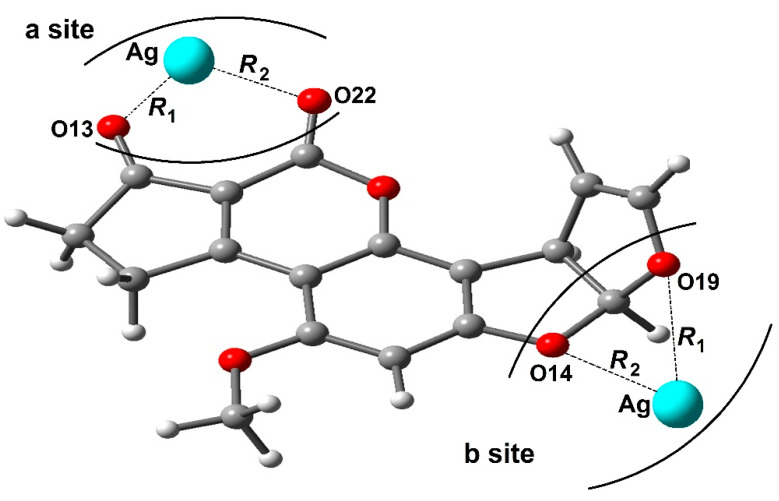
Scheme of adsorption site of AFB_1_ and AFB_1_-Ag complex. Adapted from Ref. [51].

**Figure 13 toxins-15-00135-f013:**
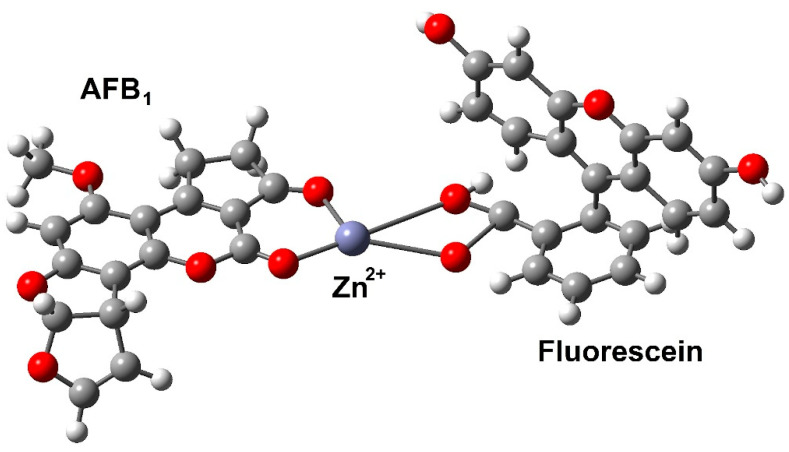
Ternary complex of Zn (II)–fluorescein–AFB_1_. Adapted from Ref. [52].

**Figure 14 toxins-15-00135-f014:**
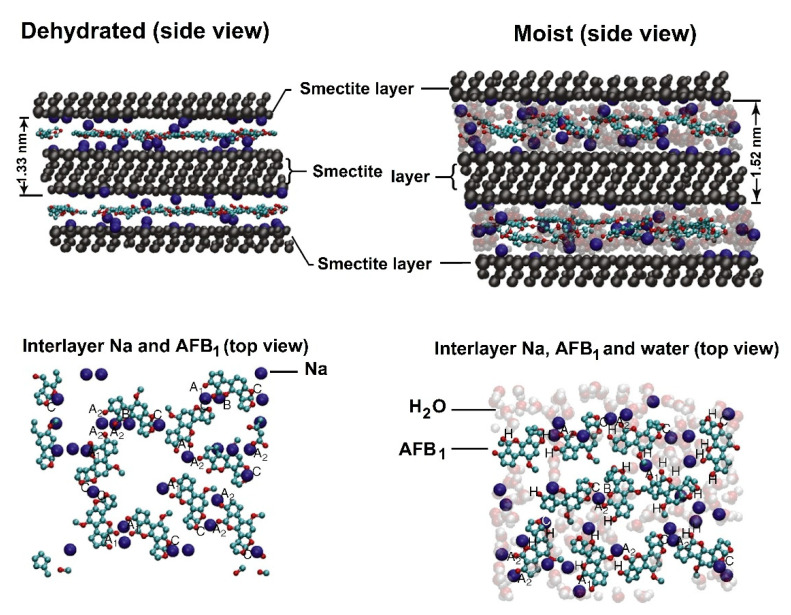
Optimized structures from molecular docking of AFB_1_–Na^+^–smectite complexes. Adapted from Ref. [53].

**Figure 15 toxins-15-00135-f015:**
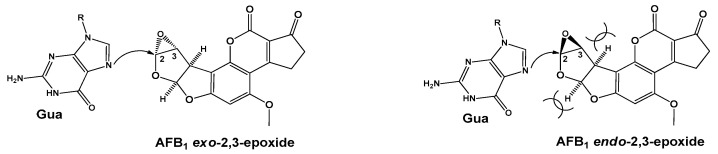
Higher reactivity of AFB_1_ *exo*-2,3-epoxide compared to AFB_1_ *endo*-2,3-epoxide. Adapted from Ref. [58].

**Figure 16 toxins-15-00135-f016:**
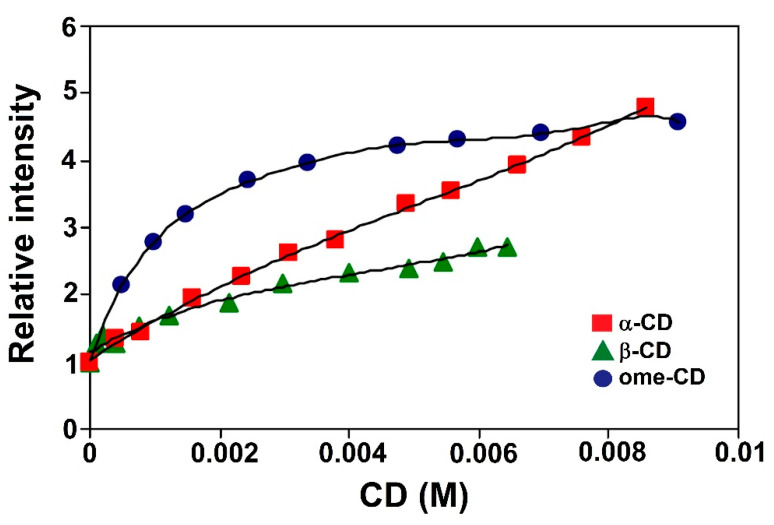
Relative fluorescence intensities of 22 nM AFB_1_ at different concentrations. Adapted from Ref. [59].

**Figure 17 toxins-15-00135-f017:**
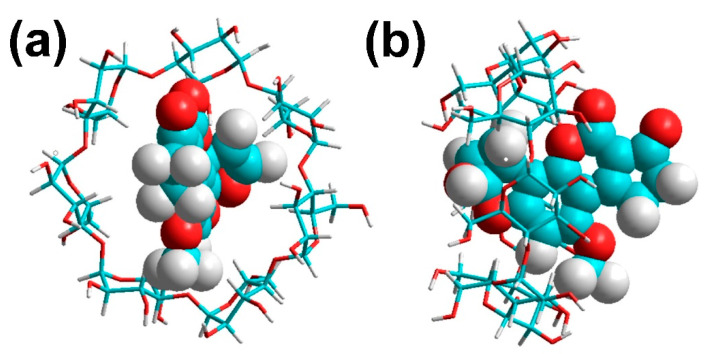
Relative host-guest geometry corresponding to the minimum of the energy of the AFB_1_–CD complex, (**a**) top secondary rim view and (**b**) side view. Adapted from Ref. [59].

**Figure 18 toxins-15-00135-f018:**
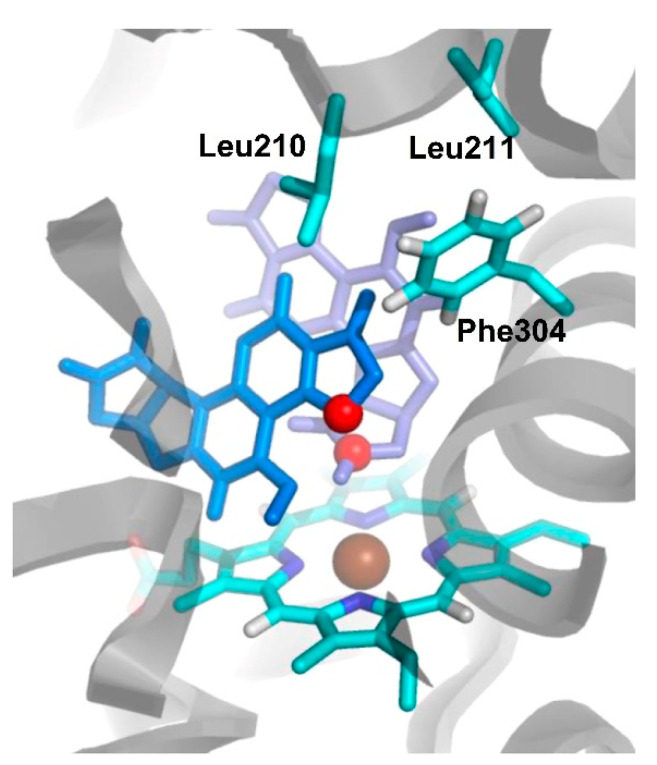
Binding mode of doubly ligated AFB_1_-CYP3A4 complex obtained by docking studies. Amino-acid residues Leu210, Leu211, Phe304, and the heme group are shown in sticks. The first AFB_1_ molecule docked is shown in violet and the second AFB_1_ molecule is shown in blue. The atoms are colored as follow: carbon in cyan, oxygen in red, nitrogen in blue, hydrogen in white, and iron in brown. Adapted from Ref. [61].

**Figure 19 toxins-15-00135-f019:**
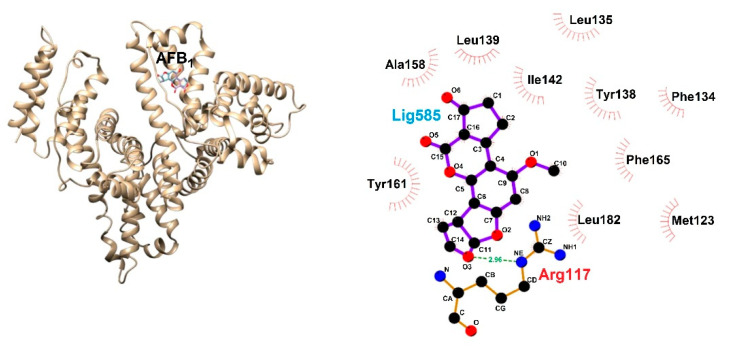
Binding mode between AFB_1_ and HAS obtained by docking studies. Adapted from Ref. [64].

**Figure 20 toxins-15-00135-f020:**
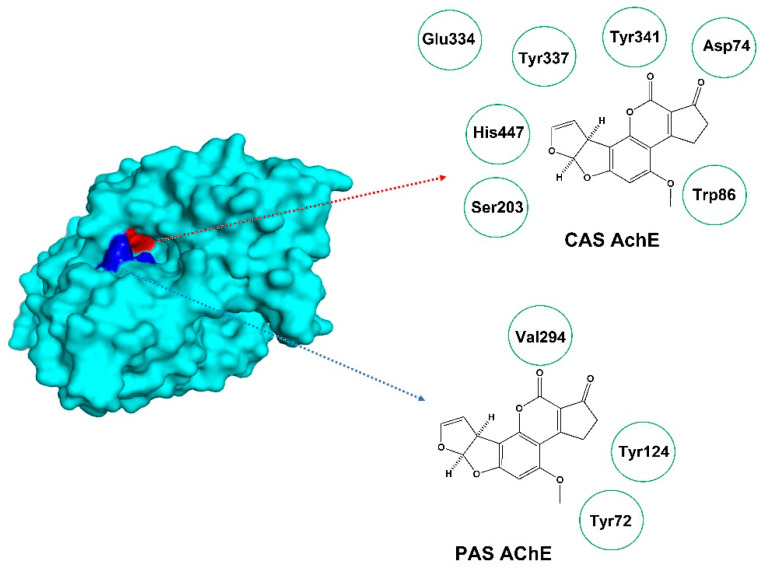
Binding mode of AFB_1_ in the PAS and CAS of AChE. It is important to highlight that the affinity exhibited by AFB_1_ for the CAS is higher than that of the PAS.

**Figure 21 toxins-15-00135-f021:**
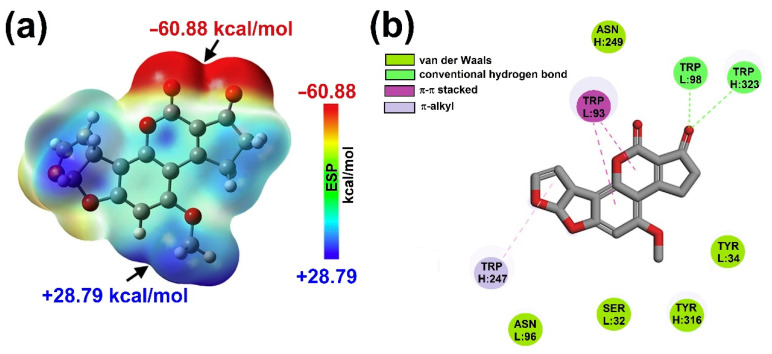
(**a**) ESP of AFB_1_, and (**b**) interactions between Fab and AFB_1_. Adapted from Ref. [68].

**Figure 22 toxins-15-00135-f022:**
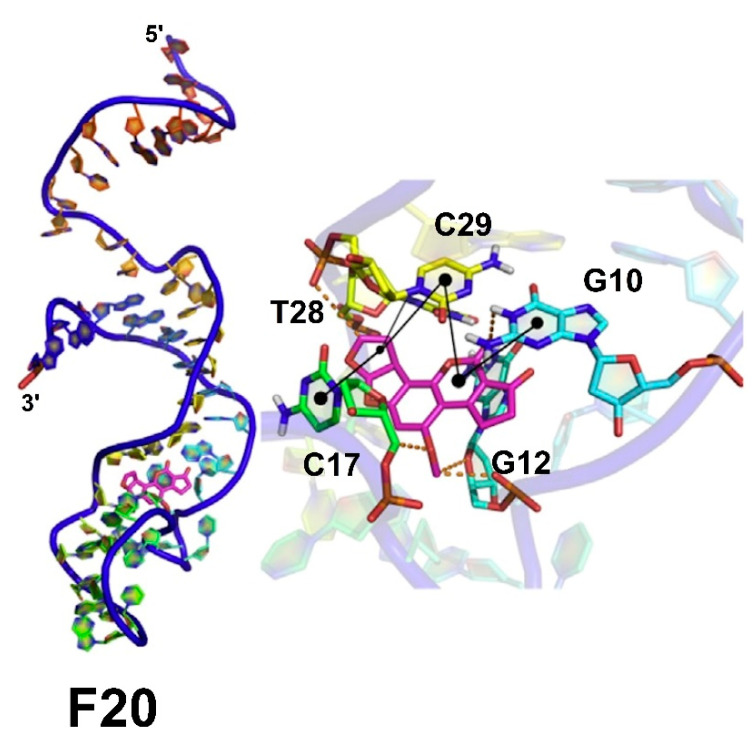
Docking result of F20-AFB_1_ complex and residues involved in binding 3D interactions. Dashes represent carbon and conventional hydrogen binding; black dots and line represent electrostatic and hydrophobic interactions. Adapted from Ref. [69].

**Table 1 toxins-15-00135-t001:** Some INDO atomic charges *q* and Wiberg bond indices *p* (in electrons, e^−^) for AFB_1_. Data from Ref. [35].

Atom	*q*A (e^−^)	Bond A-B	*p*A-B (e^−^)
O1	−0.26	O1-C2	1.03
C2	0.21	C2-C3	1.84
C3	−0.04	C3-C4	0.74
C4	−0.25	C4-C19	0.99
C5	−0.08	O1-C19	0.94
C6	0.24	C4-C5	0.99

**Table 2 toxins-15-00135-t002:** Heat of formation (*H_f_*, in kcal/mol), HOMO-LUMO energies (eV) of aflatoxins and their derivatives. Data from Ref. [36].

	AFB_1_	AFB_2_	AFG_1_	AFG_2_	AFB_1_ Epoxide	AFB_1_OH	AFB_1_ Radical
*H_f_*	−87.2	−95.7	−132.4	−177.3	−116.3	−220.0	−9.3
HOMO	−9.25	−9.44	−9.41	−9.45	−9.39	−9.36	-
LUMO	−1.48	−1.26	−1.65	−1.59	−1.53	−1.50	−6.46 ^1^

^1^ Singly occupied.

**Table 3 toxins-15-00135-t003:** Outstanding aspects: year [Ref], property, methodology (DFT or MD), and results carried out on AFB_1_.

Year [Ref]	Property	Methodology (DFT/MD)	Results
**Molecular properties of AFB_1_**			
1974 [34]	Bond order, HOMO-LUMO	Hückel molecular orbital	Relative electron affinities of AFB_1_
1985 [35]	Atomic charge, Wiberg bond indices	INDO	Higher electron density at the C2, implicates a higher stability of the carbocation in AFB_1_
1995 [36]	Geometry, HOMO-LUMO	AM1	The calculated low LUMO energies indicate that the aflatoxins are electrophilic species
2002 [37]	Energy, ^13^C NMR	Hartree–Fock/6-31G(d)	Theoretical ^13^C chemical shift matched the experimental data
2006 [38]	Geometry, atomic charges, IR	B3LYP/6-31G*	The results explain the role of methanol forms and its oxidation product in the toxic action of aflatoxin
2014 [39]	RAMAN, IR, HOMO-LUMO	B3LYP/6-311+G(d,p)	The C13 atom can serve as a position for an interaction with DNA
2018 [40]	HOMO-LUMO, excitation energy	B3LYP/6-311G	During an excitation energy decrease, the AFB_1_ molecule is unstable under external fields
**Theoretical investigations of AFB_1_**			
2000 [41]	Energy, solvation	Hartree–Fock/B3LYP/3-21G	Solvent effect implies an *endo*-attack of the nucleophiles
2002 [42]	Interaction energy	MM2	The mutagenesis could be linked to a long-standing intercalating complex and stable ligand-DNA adduct
2006 [43]	HOMO-LUMO, solvent	B3LYP/6-31+G(d,p)	The AFB_1_ suffers electronic excitation, predominantly a HOMO (-2)-LUMO transition
2008 [28]	Charge	B3LYP/6-31G(d,p)	Charge transference of the lactone ring and carbon atoms of the benzene indicate a conjugation
2010 [29]	Energy, bond order, charge, ESP, HOMO-LUMO	B3LYP/6-311+G(d,p)	The lactone ring of AFB1 was hydrolyzed, suggesting the deletion of its carcinogenic properties.
2012 [44]	Interaction energy	MD	The best interaction energies for DMC and AFB1 were obtained for allylamine and methacrylic acid
2014 [45]	Energy, solvation, dipolar moment, ESP, HOMO-LUMO	B3LYP/6-31+G(d,p) and 6-311++G(2d,2p)	TAFL and TTAFL structures can generate *exo*-2,3-epoxide and may act as carcinogenic molecules
2016 [30]	Energy	B3LYP/6-311+G(d,p), bioinformatics	The 8-chloro-9-hydroxy-AFB_1_ molecule did not present risk of mutagenicity
2017 [46]	Affinity	DFT. Docking, MD, MM/GBSA	Binding modes in CYP3A4 favors the formation of the 12α-hydroxylated and *exo*-2,3-epoxide
2020 [31]	Interaction energy	B3LYP/6-311++G(d,p)	Infrared confirmed the interaction between protonated AFB_1_ and carboxylic groups
2022 [47]	Energy	MD	Interactions between several reactive oxygen species (ROS) and AFB_1_
2022 [48]	Inactivation, mitigation	Machine learning algorithms, artificial neuronal networks	Models predicted the optimal conditions for the inactivation and mitigation of *Aspergillus parasiticus*
**Molecular interactions with inorganic compounds**			
2012 [49]	Raman	B3LYP/6-311G**	Interactions between the aflatoxins through the O atom and the Ag atom
2012 [50]	Bond length, RAMAN	B3LYP/6-311G(d,p)/LanL2DZ	The SERS enhancement factors for the AFB_1_–Ag_n_ (n = 2, 4, 6) complexes were corroborated
2014 [51]	Energy, charge, bond length, polarizability, IR, RAMAN, HOMO-LUMO	B3LYP/6-311G(d,p)Lanl2dz	The AFB1 molecule was absorbed on a silver nanoparticles by the *a* site
2020 [52]	Energies complexation, solvent	B3LYP/6-31G(d)_1dz/6-311G(d)	Dipole moments values for the obtained complexes supports the stability of the ternary complexes
2022 [33]	Interaction energy	M06-2X/6-311G(d,p), docking, MD	Complexes with two AFB_1_ molecules were most stables than those with one AFB_1_ molecule
**Molecular interactions with environmentally compounds**			
2011 [53]	Coordination energy, IR	MD	Importance of carbonyl groups in bonding AFB_1_ to smectite
2016 [54]	Coordination energy	B3LYP/6-31G(d,p)	The adsorption energy of AFB1 on the three clays was AFB_1_–smectite > AFB_1_–illite > AFB_1_–kaolinite
**Molecular interactions with biological compounds**			
1988 [55]	Interaction energy	Molecular mechanic	AFB_1_ accomplishes an adduct exclusively at guanine’s N(7) position in different DNA sequences
1989 [56]	Interaction energy	Classical empirical molecular energy potential functions	Covalent binding between AFB_1_-DNA produce conformational changes
2006 [57]	Interaction energy	Docking	The AFB_1_ interaction with β-D-glucans involves van der Waals interactions and hydrogen bonding
2007 [58]	Energy, solvation	B3LYP/HF/631G(d)/6-31+G(d,p)/6-311++G(d,p), AM1/PM3-	The stereoselectivity of this reaction exist in the aqueous solution and is further enhanced in the DNA
2007 [59]	Interaction energy	AM1, B3LYP/6-31++G**	The van der Waals interactions are responsible for the 1:1 complex formation stability
2007 [60]	HOMO-LUMO	AM1, B3LYP/6-31G*	The fluorescence of AFB_1_ is due to two configurations electronic transfer of one electron from HOMO to LUMO
2017 [61]	Interaction energy	Docking, MD	Interactions between two AFB_1_ molecules were the main source of the experimentally observed positive
2017 [1]	Interaction energy	Docking	Multi-copper-containing enzymes (laccases) can oxidize a broad range of substrates including AFB_1_
2017 [62]	Binding properties	Target fishing, docking, MD, MM/PBSA	Four proteins could be the target proteins for AFB_1_
2018 [63]	Interaction energy, affinity	MD, MM	AFB_1_ shows two binding modes: as a non-competitive inhibitor, and as a competitive inhibitor
2018 [64]	Interaction energy	Docking, MD	Thermodynamic parameters indicated that serum albumin–AFB_1_ recognition was spontaneous
2019 [65]	Interaction energy	Docking, MD	AFB_1_ metabolite could be better AChE inhibitor
2019 [66]	Interaction energy, physicochemical	Bioinformatics	AFB_1_ did not exhibit interaction with GSK-3b amino acid residues
2019 [67]	Affinity	Docking	AFB_1_ binds to the cavity of estrogen sulfotransferase
2019 [68]	ESP	B3LYP/6-311++G(d,p), docking, MD	Antibodies were recognized by AFB_1_ via van del Waals, hydrogen bonds, and π-π stacked/π-alkyls interactions
2020 [69]	Binding affinity and selectivity	Docking	The F20 aptamer carries the highest affinity and highest selectivity toward AFB_1_
2020 [71]	Binding affinity	Docking, MD	The most stable aptamers for AFB_1_ were AF_AB3 and AF_APT1
2020 [72]	Interaction energy	MD	The residue His481, present in the T1 copper, interacts with AFB_1_
2020 [73]	Interaction energy	Docking	The negative ΔG was characteristic of spontaneous and favorable interaction between albumin and AFB_1_

## Data Availability

The data reported in this study are available upon request to nicovain@yahoo.com.mx or mirruv@yahoo.com.mx.

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
