# Peer review of "Computational Studies of Aflatoxin B1 (AFB1): A Review"

_toxins, 2023, doi:10.3390/toxins15020135_

Round 1

Reviewer 1 Report

Dear authors, very interesting work, however, reading this paper raised some major concerns, please see the attached file

Reviewer 2 Report

Authors are presenting an interesting manuscript summarizing computational studies of AFB1. The manuscript is well-organized with detailed explanations and discussions. The reviewer would like to suggest a minor revision with following specific comments.

1.     In Table 1, what is the unit for charges q?

2.     In Figure 6, is it possible to determine the distances and angles between AFB1 and Ag (n=2, 4, 6)? If so, the distance and angle info can be denoted onto the figure.

3.     On page 10, line 329, authors concluded that AFL1 tautomer was the most stable both in the gas and water phases. Besides computational calculations, is there experimental evidence to support this conclusion?

4.     In Figure 8, a color meter / color bar is necessary to be added to denote the meaning of colors. 

5.     In Figure 9, can angles of Ag interaction be measured for both site a and site b?

6.     In Figure 18, there are circles, arrows, and numbers colored in red, blue, and green. Authors may want to explain them in the legend. 

7.     In Figure 21, different atoms can be colored in different colors. Residue names can be added to the plot at the corresponding positions.

Author Response

Comments and Suggestions for Authors

  • Referee: Authors are presenting an interesting manuscript summarizing computational studies of The manuscript is well-organized with detailed explanations and discussions. The reviewer would like to suggest a minor revision with following specific comments.
  • Referee: In Table 1, what is the unit for charges q?.

Us: Done, the unit has been now incorporated in Table 1; furthermore, it was also included the corresponding acronym in the text, line 155, page 4.

  • Referee: In Figure 6, is it possible to determine the distances and angles between AFB1 and Ag (n=2, 4, 6)? I so, the distance and angle info can be denoted onto the

Us: Done, the distances were included in the corresponding Figure. However, it is worth to note that related to the angles, Wu et al. did not include this data.

  • Referee: On page 10, line 329, authors concluded that AFL1 tautomer was the most stable both in the gas and water Besides computational calculations, is there experimental evidence to support this conclusion?

Us: Unfortunately, Karabulu et al. did not reported experimental data to support their conclusion; however, it is important to mention that the study of Karabulu et al. supports much other experimental research:

  • Proc. 1974, 33, 254.
  • J. Biochem. 1970, 48, 830.
  • Food Contam. 2003, 20, 1077.
  • Environ. Microbiol. 1990, 56, 1465.
  • Agric. Food Chem. 1977, 25, 437.
  • Agric. Food Chem. 1979, 27, 1339.
  • Sci. 2010, 113, 216.
  • Referee: In Figure 8, a color meter / color bar is necessary to be added to denote the meaning of

Us: Done, the color bar was included on Figure 8, please see the new version of manuscript.

  • Referee: In Figure 9, can angles of Ag interaction be measured for both site a and site b?

Us: Unfortunately, Gao et al. did not report these values; however, it is important to highlight that Gao et al. mentioned that their values agree with other works:

  • Environ. Contam. Toxicol. 2010, 59, 393.
  • Act Cryst. B 1970, 26,

  • Referee: In Figure 18, there are circles, arrows, and numbers colored in red, blue, and Authors may want to explain them in the legend

Us: Done, the corresponding legends to explain the circles, numbers, and arrows, appear in the new version of manuscript.

  • Referee: In Figure 21, different atoms can be colored in different Residue names can be added to the plot at the corresponding positions

Us: Done, the colors of atoms have been included and defined; related to the labelling of the residue names, Bren et al. did not indicate the respective residues. However, we considered convenient to use the name of amino acid residue make up the respective labelling.

Reviewer 3 Report

The review entitled "Computational studies of aflatoxin B1 (AFB1)" adresses an interesting and wide topic. The information provided in this review is quiet comprehensive.

However, I think that added value could be gained by reorganizing the manuscript and/or adding information to better introduce the topics and ease the reading.

Precisely  , I suggest to reorganize the results and discussion part of the review as follow:

-Molecular properties of AFB1(electrostactic potential potential, structure, energy,......)

-   Theorical investigations of AFB1 reactivity (degradation as depicted in figure 3 and 4 for example)

- Molecular interactions with inorganic compounds (Ag...)

- Molecular interactions with biological compounds (DNA...)

- Moleculer interactions with environmental compounds (smectite...)

And subdivide these titles for every specific target.

Further, I also suggest to start each chapter (as proposed above) by one or two paragraphs summarizing why DFT or MD studies were carried out in the specific topic : for instance why it is important to study the metabolism or the binding to albumin.... 

Finally, I strongly adding section(s) containing :

-a table summarizing typical DFT/MD studies carried out on ABF1 : context, methodoly and results 

- a discussion on the results (maybe by comparing them to what is available) and/or potential outlooks on what should be further investigated

Author Response

Comments and Suggestions for Authors

  • Referee: The review entitled "Computational studies of aflatoxin B1 (AFB1)" adresses an interesting and wide The information provided in this review is quiet comprehensive
  • Referee: However, I think that added value could be gained by reorganizing the manuscript and/or adding information to better introduce the topics and ease the
  • Referee: Precisely , I suggest to reorganize the results and discussion part of the review as follow:

-Molecular properties of AFB1(electrostactic potential potential, structure, energy,..... )

-Theorical investigations of AFB1 reactivity (degradation as depicted in figure 3 and 4 for example)

  • Molecular interactions with inorganic compounds (Ag. )
  • Molecular interactions with biological compounds (DNA. )
  • Moleculer interactions with environmental compounds (smectite. )

And subdivide these titles for every specific target.

Us: Done, the manuscript has been now reorganized, considering your commentaries, additionally, the works have been also ordered in a chronological sense; however, we kindly consider that it is not convenient subdivide each work.

  • Referee: Further, I also suggest to start each chapter (as proposed above) by one or two paragraphs summarizing why DFT or MD studies were carried out in the specific topic : for instance why it is important to study the metabolism or the binding to

Us: Done, thank you for your suggestion, some paragraphs were included highlighting the importance of DFT or MD, please see the new version of the manuscript

  • Referee: Finally, I strongly adding section(s) containing :

-a table summarizing typical DFT/MD studies carried out on ABF1 : context, methodoly and results

  • a discussion on the results (maybe by comparing them to what is available) and/or potential outlooks on what should be further investigated

Us: Done, a new Table 3, considering these commentaries lines 1590-1592, was included; additionally, the Conclusions section has been conveniently modified accompanying some other perspectives.

Reviewer 4 Report

The authors proposed a detailed review of the application of computational chemistry to aflatoxin B1, describing the different applications of theoretical calculation that occurred over time.

The review is timed and relevant for the journal, as the computational approaches have gained momentum in the food safety field. Overall, the structure proposed by the authors is sound, well-documented, and returns an exhaustive picture of the field.

Being interested in the use of computational approaches for food safety, what I've missed in this review is the critical discussion. The authors just limited their work to reporting the paper and related results following a bullet point scheme, without adding any critical comments or perspective discussion. Conclusion section is very brief, and does not add anything to the review besides saying "this is the first review...".

I must say, there are plenty of reviews popping up in the literature every week, and most of them are totally unnecessary. To set a difference, a review must provide perspective, insights, and a critical vision of a field. This is exactly what I miss from this review.

Therefore, in my view, before being considered for publication, the authors should deeply revise the text in order to make it more critical and return an in-depth commentary of the pos&cons of CC approaches.

Round 2

Reviewer 1 Report

Dear authors, the paper improved a lot

Reviewer 3 Report

The authors have revised their manuscript in depth and folowing suggestions to enhance its content. I recommand the article for publication in Toxins

Reviewer 4 Report

I do not have any comment.